



**Sources and Transformations of Anthropogenic Nitrogen along an Urban River-**
**Estuarine Continuum**
Michael J. Pennino*[1], Sujay S. Kaushal[2], Sudhir Murthy[3], Joel Blomquist[4], Jeff
Cornwell[5], and Lora Harris[6]
[1]Department of Civil and Environmental Engineering, Princeton University, Princeton,
NJ, USA.
[2]Department of Geology and Earth Systems Science Interdisciplinary Center, University
of Maryland, College Park, MD, USA.
[3]DC Water, Washington, DC, USA.
[4]U.S. Geological Survey, Baltimore MD, USA.
[5]Center for Environmental Science, University of Maryland Horn Point Laboratory,
Cambridge, MD, USA.
[6]Center for Environmental Science, University of Maryland Chesapeake Biological
Laboratory, Solomons MD, USA.
*Corresponding author email: Michael.pennino@gmail.com



**Abstract**
Urbanization has altered the fate and transport of anthropogenic nitrogen (N) in rivers
and estuaries globally.  This study evaluates the capacity of an urbanizing river-estuarine
continuum to transform N inputs from the world's largest advanced (e.g. phosphorus and
biological N removal) wastewater treatment facility.  Effluent samples and surface water
were collected monthly along the Potomac River Estuary from Washington D.C. to the
Chesapeake Bay over 150 km.  In conjunction with box model mass balances, nitrate
stable isotopes and mixing models were used to trace the fate of urban wastewater nitrate.
Nitrate concentrations and $\delta^{15}$N-NO$_3^-$ values were higher down-estuary from the Blue
Plains wastewater outfall in Washington D.C. (2.25±0.62 mg/l and 25.7±2.9‰,
respectively) compared to upper-estuary concentrations (1.0±0.2 mg/l and 9.3±1.4‰,
respectively).  Nitrate concentration then decreased rapidly within 30 km down-estuary
(to 0.8±0.2 mg/l) corresponding with an increase in organic nitrogen and dissolved
organic carbon, suggesting biotic uptake and organic transformation.  TN loads declined
down-estuary (from an annual average of 48,000±5,000 kg/day at the sewage treatment
plant outfall to 23,000±13,000 kg/day at the estuary mouth), with the greatest percentage
decrease during summer and fall.  4–71% of urban wastewater TN inputs were exported
to the Chesapeake Bay, with the greatest contribution of wastewater TN loads during the
spring.  Our results suggest that biological transformations along the urban river-estuary
continuum can significantly transform wastewater N inputs from major cities globally,
but more work is necessary to evaluate the potential of organic nitrogen to contribute to
eutrophication and hypoxia.





**Key Words**
Estuary, Mass Balance, Mixing Model, Nitrate Isotopes, Source Tracking, Wastewater

## 49 1 Introduction

Urbanization and agriculture have greatly increased the exports of nitrogen from
coastal rivers and estuaries globally, contributing to eutrophication, hypoxia, harmful
algal blooms, and fish kills (e.g. Aitkenhead-Peterson et al., 2009; Kaushal et al., 2014b;
Nixon et al., 1996; Petrone, 2010; Vitousek et al., 1997). Despite billions of dollars spent
on regulatory and technological improvements for wastewater treatment plants (WWTPs)
and agricultural and urban stormwater runoff (e.g. US-EPA, 1972, 2009, 2011), many
coastal waters are still impaired. Also, there are major questions regarding how far urban
sources of N (wastewater and stormwater runoff) are transmitted along tidal river-
estuarine networks to N-sensitive coastal receiving waters. This study evaluates the
capacity of a major river-estuarine system to transform and attenuate N inputs from the
world's largest advanced (e.g. phosphorus and biological nitrogen removal) wastewater
treatment plant (Blue Plains) before being transported down-estuary to the Chesapeake
Bay. We used a combination of stable isotope and box model mass balance approaches
to track the fate and transport of anthropogenic nitrogen across space and time.
In addition to urban and agricultural inputs, altered river-estuarine hydrology can
contribute to higher exports of N. Jordan et al. (2003) found that annual water discharge
increased as the proportion of developed land in a coastal watershed increased. Higher
flows, typically during winter and spring months, have also been associated with higher
N loads in coastal river-estuaries (Boynton et al., 2008). Furthermore, regional climate



variability amplifies pulses of nutrients and other contaminants in rivers (Easterling et al.,
2000; IPCC, 2007; Kaushal et al., 2010b; Saunders and Lea, 2008) and alters the biotic
transformation of N due to changes in hydrologic residence times (Hopkinson and
Vallino, 1995; Kaushal et al., 2014b; Wiegert and Penaslado, 1995).  For example, high
flow periods related to storms can induce stratification and impact salinity regimes
(Boesch et al., 2001), which affects nutrient biogeochemistry like ammonium and
phosphate concentrations (Jordan et al., 2008).  An improved understanding of the
longitudinal assimilatory capacity for nitrogen by large river-estuarine systems across
different flow regimes is needed for guiding effective coastal river and estuarine
management strategies.
One critical and innovative approach to effectively manage coastal nutrient
pollution is to 1) track the relative contributions of N export from different sources within
the watershed and 2) understand the potential for longitudinal transformation within
coastal rivers and estuaries.  Recent studies using stable isotopes (Kaushal et al., 2011;
Kendall et al., 2007; Oczkowski et al., 2008; Wankel et al., 2006) have shown that these
methods can be helpful in elucidating sources and transformations of nitrogen.  However,
these studies are typically conducted at relatively smaller spatial scales and without
coupling to mass balance approaches over both time and space.
Here, we combine isotope and mass balance approaches to track sources and
transformations of urban wastewater inputs to Chesapeake Bay over space and time
across an urban river-estuary continuum spanning over 150 km.  The space-time
continuum approach has previously been used in studying fate and transport of carbon
and nitrogen in urban watersheds (Kaushal and Belt, 2012; Kaushal et al., 2014c), and



here we explore extending it to river and estuarine ecosystems.  Our overarching
questions were: 1) how does the importance of point *vs.* non-point sources of N shift
along a tidal and stratified urban river-estuary continuum across space and time?  2) What
is the capacity of an urban river-estuary continuum to transform or assimilate
anthropogenic N inputs?  3) How are transport and transformations of N affected by
differences in season or hydrology?  An improved understanding of how sources and
transformations of N change along the urban river-estuarine continuum over space and
time can inform management decisions regarding N source reductions along urbanizing
coastal watersheds (e.g. Boesch et al., 2001; Kaushal and Belt, 2012; Paerl et al., 2006).
## 2    Methods
### 2.1    Site Description

This study is focused on the tidal Potomac River Estuary, which includes the

section of the river from Washington D.C. to its confluence with the Chesapeake Bay
(Fig. 1).  The Potomac River Estuary begins as tidal freshwater, becoming oligohaline
~30-50 km below Washington D.C., and mesohaline at its mouth approximately 160 km
below Washington D.C. (Jaworski et al., 1992).  The Potomac River Estuary can be
seasonally stratified (Hamdan and Jonas, 2006), especially in the southern portion of the
system where intruding, saline bottom water from the main stem of the Chesapeake Bay
leads to density driven estuarine circulation patterns (Elliott, 1976, 1978; Pritchard,
1956).  Mixing is most evident at the estuarine turbidity maximum (Hamdan and Jonas,
2006), ~60-80 km below Washington D.C., and the water column is generally well mixed
above the estuarine turbidity maximum zone in the tidal fresh and oligohaline regions of



the estuary (Crump and Baross, 1996; Sanford et al., 2001).
The watershed draining to the Potomac River Estuary is classified as 58% forested,
23% agricultural, and 17% urban, based on Maryland Department of Planning data for
2002 (Karrh et al., 2007a).  Based on the Chesapeake Bay Program (CBP) Model it was
estimated that during 2005 total inputs of nitrogen were 33% from agriculture, 20% from
urban (e.g. stormwater runoff and leaky sewers), 19% from point sources (wastewater
treatment plants and industrial releases), 11% from forest, 10% from septic, 6 % from
mixed open land, and 1 % from atmospheric deposition to water (Karrh et al., 2007b).
The CBP model is developed using long-term monitoring data and the non-point loads
are estimated from a variety of sources including land cover and agriculture records
(Karrh et al., 2007b).  The Potomac River Estuary also receives N inputs from Blue
Plains wastewater treatment plant, located in Washington, D.C.  Blue Plains currently
discharges 2.3 mg/L of $NO_3^-$ and 3.7 mg/L of TN, on average, and exports loads of
approximately 2,300 kg/day of $NO_3^-$ and 3,900 kg of TN.  Overall, Blue Plains treats and
discharges 280 million gallons per day (mgd), almost 5% of Potomac River's annual
discharge.  In the past several decades, Blue Plains has undergone several technological
improvements with phosphorus removal in the 1980s and enhanced N removal beginning
in the year 2000.  Since the implementation of advanced wastewater treatment
technologies at Blue Plains, there has been a significant decrease ($p < 0.01$) in the
concentration of nitrate in effluent discharge, from an average of $7.2 \pm 0.3$ mg/L before
the year 2000 (years 1998 and 1999) to an average of $4.1 \pm 0.4$ mg/L after 2000 (years
2001 to 2008).



**2.2    Analysis of long-term spatial and temporal water chemistry data**

Surface and bottom water N and carbon data collected by the Maryland

Department of Natural Resources (DNR) and accessed through the Chesapeake Bay
Program's data hub website (Chesapeake Bay Program, 2013) was used to look at
historical (1984 to 2012) monthly nutrient concentrations from stations located
longitudinally along the Potomac River Estuary (Fig. 1).  These data were used to look at
the spatial and temporal trends for dissolved and particulate forms of N and dissolved
organic carbon (DOC) in the Potomac River Estuary prior to and during this study.

**2.3    Water Sampling**

Water samples along the Potomac River estuary were collected monthly for one

year from April 2010 to May 2011; from 12 km to 160 km below the Blue Plains
wastewater treatment plant (See Fig. 1).  Water was collected from the surface (top 0.5
m) and bottom water depths.  Additionally, surface water samplings from 6 km above to
12 km below the Blue Plains wastewater treatment plant effluent outfall were collected
seasonally during this time (Fig. 1).  Water temperature and salinity was also measured
during each water sampling.

**2.4    Nitrate $\delta^{15}$N and $\delta^{18}$O Isotope Analysis**

Surface samples for $\delta^{15}$N-NO$_3^-$ and $\delta^{18}$O-NO$_3^-$ isotopes of dissolved nitrate were

filtered (0.45 μm), frozen, and shipped to the UC Davis Stable Isotope Facility (SIF) for
analysis.  The isotope composition of nitrate was measured following the denitrifier
method (Casciotti et al., 2002; Sigman et al., 2001).  In brief, denitrifying bacteria are



used to convert nitrate in samples to $N_2O$ gas, which is collected and sent through a mass
spectrometer for determination of the stable isotopic ratios for N and O of nitrate ($^{15}N/^{14}N$
and $^{18}O/^{16}O$).  Values for $\delta^{15}N$-$NO_3^-$ and $\delta^{18}O$-$NO_3^-$ are reported as per mil (‰) relative
to atmospheric $N_2$ ($\delta^{15}N$) or Vienna Standard Mean Ocean Water (VSMOW) ($\delta^{18}O$),
according to $\delta^{15}N$ or $\delta^{18}O$ (‰) = [(R)sample / (R)standard - 1] × 1000, where R denotes
the ratio of the heavy to light isotope ($^{15}N/^{14}N$ or $^{18}O/^{16}O$).  For data correction and
calibration UC Davis SIF uses calibration nitrate standards (USGS 32, USGS 34, and
USGS 35) supplied by NIST (National Institute of Standards and Technology,
Gaithersburg, MD).  The long-term standard deviation for nitrate isotope samples at UC
Davis SIF is 0.4 ‰ for $\delta^{15}N$-$NO_3^-$ and 0.5 ‰ for $\delta^{18}O$-$NO_3^-$.  Previous studies (Kaushal
et al., 2011; Kendall et al., 2007) indicate that the relative amounts of $\delta^{15}N$-$NO_3^-$ and
$\delta^{18}O$-$NO_3^-$ can be used to determine specific sources of nitrate (i.e. fertilizer, nitrification,
atmospheric, or sewage derived nitrate).

It should be noted that while the denitrifier method converts sample $NO_3^-$ and

$NO_2^-$ to $N_2O$ gas, in marine systems, $NO_2^-$ has been shown to complicate interpretations
of the N and O isotopes of $NO_3^-$ if it remains unaccounted for (e.g. Fawcett et al., 2015;
Marconi et al., 2015; Rafter et al., 2013; Smart et al., 2015).  This is partially because
during the reduction of $NO_3^-$ and $NO_2^-$ to $N_2O$ by the denitrifiers, the O isotope effects
are different (and thus need to be corrected for).  In addition, the $\delta15N$ of $NO_2^-$ can be
extremely different from that of $NO_3^-$, potentially further complicating interpretation of
the data.



**2.5     Nitrate Isotope Mixing Model**

To distinguish between the different potential nitrate sources we used a Bayesian

isotope mixing model (Parnell et al., 2010; Parnell et al., 2013; Xue et al., 2012; Yang
and Toor, 2016).  For the Bayesian isotope mixing model, the Stable Isotope Analysis in
R (SIAR) package was used to determine the fraction of nitrate in each sample from four
different sources: wastewater, atmospheric, nitrification, and nitrate fertilizer (Parnell et
al., 2010; Parnell et al., 2013; Xue et al., 2012; Yang and Toor, 2016).  The SIAR mixing
model is able to incorporate uncertainty in nitrate source estimates based on the
uncertainty in the nitrate source endmembers (see below) (Parnell et al., 2010; Parnell et
al., 2013; Xue et al., 2012; Yang and Toor, 2016).

Nitrate source end-member values, for $\delta^{15}$N-NO$_3^-$ and $\delta^{18}$O-NO$_3^-$ were obtained

from the literature, except wastewater nitrate, which was obtained from this study.  The
end-member values for $\delta^{15}$N-NO$_3^-$ and $\delta^{18}$O-NO$_3^-$ were -10.3±1.7 and 10.1±1.5,
respectively for nitrate from nitrification (Mayer et al., 2001), 0±3 and 22±3, respectively
for NO$_3^-$ fertilizer (Mayer et al., 2002), and 3±3 and 69±5, respectively for atmospheric
nitrate (Burns and Kendall, 2002; Divers et al., 2014).  The wastewater $\delta^{15}$N-NO$_3^-$ and
$\delta^{18}$O-NO$_3^-$ end-member values (31.5±7.8 and 11±4.5, respectively) were based on
averaging the effluent nitrate isotope values measured monthly from Blue Plains during
the study period.  The nitrification source represents NO$_3^-$ from nitrification in the water
as well as nitrification of ammonia fertilizer in the watershed.  The fertilizer source
represents synthetically produced NO$_3^-$ fertilizer, not the more common ammonia
fertilizer.



Due to the variability in nitrate source endmembers, the mixing model was used
primarily for illustrative purposes and should be viewed with caution (particularly with
regard to identifying other sources besides wastewater).  For example, there is high
variability in the nitrification source endmembers because nitrate from nitrification can
come from ammonia fertilizer, manure fertilizer, particulate organic matter within the
water column, etc.  The nitrate from nitrification will therefor carry a range of nitrate
isotope values reflecting its original source (Kendall et al., 2007).  Additionally, because
denitrification is known to cause the increase in $\delta^{15}$N-NO$_3^-$ and $\delta^{18}$O-NO$_3^-$ values through
isotopic fractionation in approximately a 2:1 relationship (Divers et al., 2014; Kendall et
al., 2007), this isotopic enrichment can complicate the identification of wastewater
nitrate.  As a result, water samples with increased wastewater nitrate, based on the mixing
model, may also suggest denitrification has played a role in the isotopic levels of the
sample nitrate.

**2.6     Salinity *vs*. Nitrate Concentration and Isotope Mixing Plots**
An additional method using plots of salinity *vs*. NO$_3^-$ concentration or NO$_3^-$
isotopes was used to assess whether there is conservative mixing (dilution), or mixing
with additional NO$_3^-$ sources down-estuary, or losses of NO$_3^-$ through biotic uptake or
denitrification (Middelburg and Nieuwenhuize, 2001; Wankel et al., 2006).  Mixing line
equations for NO$_3^-$ concentrations were based on equations 1-3 from Middelburg and
Nieuwenhuize (2001) and isotopes mixing lines were based on equation 4 from
Middelburg and Nieuwenhuize (2001).  The mixing line equations and endmember
values used for salinity and nitrate isotopes are provided in supporting information (Table



S2).  Based on those equations, the salinity *vs*. $NO_3^-$ concentration mixing lines are linear,
while the mixing lines for $NO_3^-$ isotopes are non-linear (Middelburg and Nieuwenhuize,
2001).  Wankel et al. (2006) suggests that when nutrient concentrations fall above the
mixing line this indicates an additional source to raise the concentrations, while
concentrations that fall below the mixing line indicate there is a nutrient sink (e.g.,
denitrification, assimilation, etc.).  For nitrate isotopes, when the $\delta^{15}N\text{-}NO_3^-$ and $\delta^{18}O\text{-}$
$NO_3^-$ values fall above this mixing line, this could indicate an additional source or the
fractionation of nitrate from assimilation or denitrification that would increase the heavy
isotope levels, while isotope values below the mixing line could indicate an additional
source of nitrate with lighter isotope values, such as from nitrification or fertilizer sources
(Wankel et al., 2006).

**2.7    Estuarine Nitrogen Net Fluxes**

A box model was used to estimate net fluxes of TN, $NO_3^-$, and nitrate isotope

loads along the Potomac River Estuary using methods modified from Officer (1980),
Boynton et al. (1995), Hagy et al. (2000), and Testa et al. (2008), which are widely used
methods for tracking nutrient fluxes in estuaries between different salinity zones.  First,
the Potomac Estuary was divided into 6 boxes in order to accommodate adequate
sampling stations per box, and to evaluate net fluxes at key locations along the estuarine
gradient (Fig. 2).  Next, due to the Potomac Estuary having a semi-diurnal tidal cycle,
where there is movement back and forth across boundaries of the box model, mean
monthly freshwater discharge inputs to the first box (USGS, 2014) and interpolated
salinity values (measured monthly from surface and bottom waters throughout the



system) were used to calculate advective and diffusive exchanges of water and salt
between adjacent boxes.  Salt balances were then used to compute net exchanges at the
boundaries of the six model boxes, similar to previous estuarine box model studies (e.g.
Boynton et al., 1995; Hagy et al., 2000).  Average monthly TN, $NO_3^-$ and $NO_3^-$ isotope
concentrations (collected from the surface and bottom water at each station, except for
$NO_3^-$ isotopes, which were collected from the surface only) were multiplied by net
estimated exchange values at the box boundaries and summed to calculate the N load
leaving or entering each box.  In order to calculate the loads for $NO_3^-$ isotopes, the $\delta^{15}N$-
$NO_3^-$ and $\delta^{18}O$-$NO_3^-$ values in per mil (‰) were converted to concentrations (µg/L) by
multiplying the $NO_3^-$ concentration of the sample by R, the ratio of the heavy to light
isotope ($^{15}N/^{14}N$ or $^{18}O/^{16}O$).  Fluxes were estimated for each month during the sampling
period and then averaged to find seasonal estimates of N fluxes for the Potomac.  The
box model results were used to compute: (1) the total inputs of N, (2) the % inputs of
loads from Blue Plains, (3) the net export of N to the Chesapeake Bay, (4) the % of Blue
Plains inputs that are exported, (5) the net loss in loads along the estuary, and (6) the
contribution of N loads from the Chesapeake Bay through tidal inflow.

To account for uncertainty in monthly load estimates, error propagation was used

for each of the hydrologic and nutrient inputs to the model.  For example, the error in
discharge data came from averaging the mean daily discharge for each month, the error in
water concentrations came from averaging the surface and bottom water concentrations,
and the error in N from atmospheric deposition came from averaging the weakly
deposition data for each month.  These uncertainties in the inputs to the box model were



then propagated for each of the box model calculations, similar to Filoso and Palmer

(2011).

Inputs to the box model include, total monthly precipitation data based on

averaging data from three stations along the Potomac Estuary (Precipitation data is from
the NOAA National Centers for Environmental Information, Climate Data Online),
monthly estimates of atmospheric deposition for $NH_4^+$, $NO_3^-$, and DIN (obtained from the
National Atmospheric Deposition Program / National Trends Network), $NO_3^-$
concentrations and isotope levels in atmospheric deposition (from Buda and DeWalle,
2009, for the nearby central Pennsylvania region for the year 2005, which was a similar
year hydrologically (as described below)), N inputs from the land (from Chesapeake Bay
model output from 2005), surface and bottom water nutrient and salinity concentrations
(from MD DNR), and inputs from the Blue Plains wastewater treatment plant.  Also,
while there are no USGS gages located along the Potomac Estuary, there is one USGS
gage (USGS 01646580) located directly above the Estuary, above the fall line (the
location where the hydryodynamics of the river cease being tidally influenced) and this
gage was used to account for freshwater inputs into the first box.  The model also takes
into account water temperature and evaporation.

For the box model it is assumed that the 14 other WWTPs further down-estuary

have little effect on the nitrate signal because their combined TN load is 32% of the TN
from Blue Plains and for the other reasons described below.  While $\delta^{15}$N-$NO_3^-$ and $\delta^{18}$O-
$NO_3^-$ isotope values were not measured directly for the 14 other down-estuary wastewater
treatment plants, based on the literature, the values of these isotopes are typically lower
(~10‰ for $\delta^{15}$N-$NO_3^-$ and ~0 for $\delta^{18}$O-$NO_3^-$) compared to 31.5‰ for $\delta^{15}$N-$NO_3^-$ and



11‰ $\delta^{18}$O-NO$_3^-$ for Blue Plains (Kendall et al., 2007; Wang et al., 2013; Wankel et al.,
2006).  As a result, we expected the other WWTPs to have a similar or an even less
pronounced wastewater isotope signal compared to Blue Plains, which has biological
nitrogen removal (i.e. denitrification is promoted within the WWTP), elevating the $\delta^{15}$N-
NO$_3^-$ and $\delta^{18}$O-NO$_3^-$ isotope values at Blue Plains more (Kendall et al., 2007).
Consequently, the estimated nitrate loads down-estuary incorporate Blue Plains and
nitrate inputs from the other WWTPs, and are considered conservative estimates because
the additional WWTPs only add to the TN loads and NO$_3^-$ isotope signals and thus lessen
the decline in loads or isotope values down estuary.

A second assumption was made for the box model related to estuarine mixing.

Although portions of the lower estuary can be seasonally stratified, we assumed each box
to be well mixed vertically as no bottom water isotope values were available to constrain
a 2-layer box model.  This assumption is supported by other bottom water data that is
available and by samples taken along the width of the estuary.  For example, we have
conducted the box model and other analyses with and without bottom water isotope data
and found minimal change in results (Fig. S1, see below).  Our measurements of various
biogeochemical signatures at the station close to the estuarine turbidity maximum
suggests that there is intense mixing at this site, and prior studies have documented
extensive mixing in the freshwater tidal portion of the system (Elliott, 1976, 1978;
Pritchard, 1956).  Also, it can be assumed that because wastewater effluent inputs are
freshwater, much of the effluent plume would likely not sink in the more dense estuarine
waters moving up from the bay.  Additionally, our box model estimates of net fluxes was
compared to a complex, 3 dimensional hydrodynamic model (described below) that



incorporates stratification, and this comparison provided support for the low impact of
assuming mixing in our approach.

While seasonal stratification has been found close to the mouth of the of the

Potomac estuary (Hamdan and Jonas, 2006), using documented nitrate bottom water
isotope values from near the mouth of the estuary (Horrigan et al., 1990) we calculate
that incorporating bottom water isotope values would have a minimal impact on the flux
estimates of our box model, particularly when not including spring 2011 (Fig. S1). But
when including spring 2011, and using the reported values of 10‰ for bottom water
$\delta^{15}N$-$NO_3^-$, based on Horrigan et al. (1990), in Boxes 5 and 6 where stratification is most
likely, our estimates for the flux of $\delta^{15}N$-$NO_3^-$ from these boxes increases by 20% on
average, and the net loss in load from box 1 to box 6 increases by 12% on average. This
indicates that our estimates are conservative because by not using bottom water we
estimate a smaller net loss in $\delta^{15}N$-$NO_3^-$ (Fig. S1).

For the box model we also assumed the estuary to be well mixed laterally. In

terms of potential variability for samples taken at different locations along the width of
the estuary, there was found for surface water samples, on average, a 6±3% difference in
$\delta^{15}N$-$NO_3^-$, a 7±3% difference in $\delta^{18}O$-$NO_3^-$, a 24±8% difference in $NO_3^-$, and a 15±3%
difference in TN (based on samplings that were done at two or more locations along the
same longitudinal transect at approximately the same distance down-estuary, but at
different locations horizontally at that location). Based on this, the nitrate isotopes values
and $NO_3^-$ and TN concentrations appear to show that the estuary is fairly well mixed
laterally.





To assess the accuracy of the box model assumptions and results, estimated net
fluxes of total N were compared to simulation output from the Chesapeake Bay Water
Quality Model.  This model was developed by the U.S. EPA to aid in efforts to set
TMDLs for the Chesapeake Bay (Cerco et al., 2010), and combines a 3-D hydrodynamic
model (CH3D) with a water quality model (CE-QUAL-ICM).  Simulation output data
were available for 1996, 2002, and 2005.  We selected a simulation year (2005) because
it had similar river discharge conditions to 2010, and compared modeled net fluxes of TN
at three boundary locations to estimates at the same (or nearby) box model boundaries.

**2.8    Statistical Analyses**
Statistical analyses were performed using the statistical package R (R
Development Core Team, 2013).  Linear regression was used to test for significant
changes in stream chemistry and nitrate isotope data with distance down estuary.
Repeated measures analysis of variance (ANOVA) was used to test for seasonal
differences in nitrate isotopes trends with distance.
**3    Results**
**3.1    Spatial and Temporal Trends in N Concentrations**
Longitudinal patterns of dissolved inorganic nitrogen (DIN) in the lower
Potomac River showed an increase in concentrations near and directly below the Blue
Plains wastewater treatment plant and then a steady decline in concentrations down to the
Chesapeake Bay (Fig. 3a).  The implementation of tertiary treatment in 2000 coincided
with a significant drop in annual average DIN concentration directly down-estuary (from



$1.7 \pm 0.02$ to $1.3 \pm 0.01$ mg/l, $p < 0.05$) (Fig. 3a), when comparing years directly prior
(1997-1999) and the years directly after 2000 (2001-2005).  However, the impact of the
wastewater treatment plant improvements on reducing longitudinal patterns of DIN was
only apparent for the first 30 km down-estuary.  After this, both the pre- and post-2000
DIN concentrations overlapped (Fig. 3a).  As DIN decreased longitudinally down-estuary
of the wastewater treatment plant, there was also a small, but significant increase in total
organic nitrogen (TON) after the year 2000 ($p < 0.01$, Fig. 3a), not including the last
sample near the mouth of the estuary, which is likely influenced by tidal inflow.

There were seasonal variations in DIN concentrations along the Potomac River

Estuary with the greatest concentrations in the winter and spring (Fig. 3b).  There is also
a steeper decline in DIN with distance during fall, winter, and summer compared to the
spring ($p < 0.05$, Fig. 3b).  The average molar ratio of DIN to $PO_4^{-3}$ (N:P ratio) showed
an initial increase, then a decrease as estuarine salinity started to increase (Fig. 3c).
During the summer and fall, the N:P ratio fell below the Redfield ratio (16:1, the atomic
ratio of nitrogen and phosphorus found in oceans and phytoplankton), around 40 km
down-estuary and stayed below 16, which indicated a shift from P to N limitation.
During the winter and spring, the N:P ratio never fell below 16 and increased steadily
after 50 km down-estuary (Fig. 3c).  There was also a significant negative relationship
between $NO_3^-$ and DOC concentration during the study period ($p < 0.01$, Fig. 4).

**3.2    Spatial and Seasonal Trends in $NO_3^-$ Isotopes and Sources**

During each season, except spring, $\delta^{15}N$-$NO_3^-$ values increased sharply at the

Blue Plains outfall, from $9.3 \pm 1.4$ ‰ up-estuary to $25.7 \pm 2.9$ ‰ at the outfall ($p < 0.05$),



and then rapidly decreased within 2 km down-estuary to $15.7 \pm 2.2$ ‰ ($p < 0.05$, Fig. 5a).
During the summer and fall, the $\delta^{15}N\text{-}NO_3^-$ values showed the largest increase near the
effluent outfall (except for one very high winter value) and then a significant decrease ($p$
$< 0.05$) with distance down-estuary.  There was also a slight increase in $\delta^{15}N\text{-}NO_3^-$ and
$\delta^{18}O\text{-}NO_3^-$ values from 1 to 6 km down-estuary (Fig. 5a,b).  During the winter and spring,
the $\delta^{15}N\text{-}NO_3^-$ and $\delta^{18}O\text{-}NO_3^-$ values remained relatively constant throughout the estuary,
even near Blue Plains (Fig. 5a,b), while during the summer and fall the $\delta^{15}N\text{-}NO_3^-$ and
$\delta^{18}O\text{-}NO_3^-$ values steadily declined after 6-10 km down-estuary (Fig. 5a,b).  At the mouth
of the estuary, the $\delta^{15}N\text{-}NO_3^-$ values for all seasons were roughly equivalent (Fig. 5a).
During the summer and fall, the $\delta^{18}O\text{-}NO_3^-$ values showed a steady decrease after 12 km
down-estuary, while they increased during spring and winter (Fig. 5b).

Based on the nitrate isotope mixing model, nitrate contributions from wastewater

ranged from $80 \pm 13\%$ at the wastewater outfall to $57 \pm 11\%$ within the first 1 km down-
estuary.  Wastewater nitrate contributions then decreased to $44 \pm 14\%$ at the confluence
of the Potomac River Estuary with Chesapeake Bay (Fig. 5c).  Nitrate from nitrification
(of N from upriver manure or ammonia fertilizer and also Blue Plains wastewater N)
increased from $13 \pm 12\%$ at the wastewater outfall to $29 \pm 22\%$ at the confluence of the
Potomac River Estuary with Chesapeake Bay (Fig. 5c).  Nitrate from fertilizer increased
from $6 \pm 6\%$ at the wastewater outfall to $22 \pm 22\%$ at the confluence of the Potomac
River Estuary with Chesapeake Bay (Fig. 5c).  Nitrate from atmospheric deposition
changed little along the Potomac Estuary from $1 \pm 1$ at the wastewater outfall to $5 \pm 5$ at
the confluence with the Chesapeake Bay (Fig. 5c).  At the last two sampling stations near



the mouth of the Potomac River Estuary, $NO_3^-$ from fertilizer showed an increase, while
$NO_3^-$ from nitrification showed a corresponding decline (Fig. 5c).

**3.3    $\delta^{15}N\text{-}NO_3^-$ and $\delta^{18}O\text{-}NO_3^-$, $NO_3^-$ Concentration, and Salinity Relationships**

The Blue Plains effluent and Potomac River samples within 20 km downriver of
the wastewater treatment plant showed a significant positive relationship between $\delta^{15}N$-
$NO_3^-$ and $\delta^{18}O\text{-}NO_3^-$ ($p < 0.05$) (Fig. 6a). When denitrification and biotic uptake occurs,
plotting $\delta^{15}N\text{-}NO_3^-$ *vs.* $\delta^{18}O\text{-}NO_3^-$ shows a 2:1 relationship (Kendall et al. 2007). The
Blue Plains effluent samples showed approximately a 2.4 to 1 relationship. The samples
within 20 km downriver showed a 3:1 ratio (Fig. 6a). The nitrate samples within the first
6 km showed a 2.4 to 1 relationship (Fig. 6a). There were also seasonal differences in the
relationship between $\delta^{15}N\text{-}NO_3^-$ and $\delta^{18}O\text{-}NO_3^-$ (Fig. 6b); spring, summer, and fall were
characterized by close to a 2:1 relationship between $\delta^{15}N\text{-}NO_3^-$ *vs.* $\delta^{18}O\text{-}NO_3^-$, while
winter showed a ~8:1 relationship.
Because salinity is a conservative tracer, plots of salinity *vs.* $NO_3^-$, $\delta^{15}N\text{-}NO_3^-$, and
$\delta^{18}O\text{-}NO_3^-$ can indicate effects of mixing between water at the tidal freshwater section
with water from the mesohaline section of the Potomac River Estuary. Deviations from
the mixing lines can indicate additional sources or biological transformations
(Middelburg and Nieuwenhuize, 2000; Wankel et al., 2006). Surface water $NO_3^-$
concentrations and nitrate isotopes fell on (for $\delta^{18}O\text{-}NO_3^-$) or slightly below mixing line
(for $\delta^{15}N\text{-}NO_3^-$) during the spring (Fig. 7a,b,c), which indicated mostly conservative
mixing (dilution or inputs from low $\delta^{15}N\text{-}NO_3^-$ like nitrification, see discussion below).
But during the summer and fall, the $NO_3^-$ concentration and isotope values fell well





below the mixing lines.  During the winter, the values fell both above and below the
mixing line (Fig. 7a,b,c), which indicated non-conservative mixing (please see discussion
below).

**3.4     Spatial and Seasonal Trends in N Loads**

Our comparisons of box model net exchange estimates with simulation output

provided by the Chesapeake Bay Program Eutrophication Model ("Bay Model") revealed
similar TN loads between our results and the Bay Model in the winter, spring, and fall,
with the largest differences in the models evident in the summer months at the boundary
location where tidal fresh transitions to oligohaline conditions and at the mouth of the
estuary (Table S3 and Figures 8 and 9).  Even so, these differences are smaller than a
factor of 2 for winter and spring and for most of the summer and fall, despite the
assumption of complete mixing in our box model, a good agreement considering the
simplification of hydrodynamics inherent to a box modeling approach when compared to
the highly constrained CH3D hydrodynamic modeling platform (Cerco et al., 2010).  The
Potomac estuary is well mixed along two thirds of its length, and this likely contributes to
our success in applying a single layer box model to this system.  The box model also
permitted estimates of TN loads at smaller spatial scales than the three boundaries
available from the Chesapeake Bay Program, which could enable a better interpretation
of where Blue Plains effluent was subject to transformations in the oligohaline portion of
the estuary (Fig. 8).  The caveat here is that box-modeled summer loads should be
interpreted with caution because they show the greatest differences from the CH3D
model.





Results of the box model indicated that the net load (kg/day) of TN, $NO_3^-$, and

$\delta^{15}N\text{-}NO_3^-$ decreased down-estuary during each season (Fig. 10a-c, p <0.05 for winter
and spring and p < 0.1 for summer and fall).  N loads were highest along the estuary
during spring and winter (Fig. 10), and there was a greater decline in TN loads on
average from box 1 to box 6 during winter and spring (a loss of ~27,000 ± 15,000 and
50,000 ± 52,000 kg/day, respectively) (Table 1) compared to summer and fall (a loss of
~7,000 ± 8,000 and 15,000 ± 13,000 kg/day, respectively).  However, the summer and
fall months showed a greater percent decline in TN (75 ± 75% and 112 ± 95%,
respectively) compared to winter and spring (54 ± 40 and 36 ± 43%, respectively).  The
relatively high errors are primarily from the larger uncertainty found in the last box, at the
mouth of the estuary, due to the larger size of this box and greater uncertainty in fluxes at
the mouth of the estuary; the uncertainties are much smaller further up-estuary (See Fig.
10a).  $NO_3^-$ and $\delta^{15}N\text{-}NO_3^-$ follow the same seasonal patterns as TN.  Also, winter, along
with summer and fall, showed a greater percent decline in $NO_3^-$ and $NO_3^-$ isotope loads
compared to spring (Table 1).

Using an estimated N burial rate of $7.09 \times 10^6$ kg/yr (which is an average of burial

rate estimates for the upper and lower Potomac Estuary) from Boynton et al. (1995), it
was calculated that, on an mean annual basis, burial accounts for about 77% of the loss in
TN.  Denitrification was then calculated, by difference, to account for the remaining 23%
loss in TN load.  Using a different independent method, based on the average annual
estimated denitrification rate ($2.8 \times 10^6$ kg/yr) from the upper and lower Potomac
(Boynton et al., 1995), and the box model results, it is estimated that denitrification



accounts for about 27% of the TN removal. Consequently denitrification is estimated to
account for 23 to 27% of the loss in TN load along the Potomac Estuary.

The percent contribution of TN inputs from the Blue Plains wastewater treatment

to the main stem of the Chesapeake Bay ranged from 8 to 47 % (Table 1). The
contribution was significantly lower during the winter and spring ($10 \pm 13$ and $8 \pm 1\%$,
respectively) compared to summer and fall ($38 \pm 3$ and $47 \pm 13\%$, respectively, Table 1),
when TN fluxes from all sources are relatively low. The percent contribution of Blue
Plains wastewater TN inputs, which are exported to the Chesapeake Bay ranged from <4
to 71%, and they were highest in the spring ($71 \pm 20\%$, Table 1). There were also N
inputs to the Potomac river-estuarine continuum from the Chesapeake Bay during each
season, except spring, due to higher flows (Table 1 & 2) because flow in spring was too
high to allow the inputs from the Bay that occurred in the other seasons. $NO_3^-$ and $\delta^{15}N$-
$NO_3^-$ follow the same seasonal patterns as TN, showing the greatest percentage of inputs
from Blue Plains exported during the spring.

## 4    Discussion

While coastal urbanization can have a major impact on water quality in receiving

waters, the results of this study suggest that estuaries also show a large capacity to
transform or bury anthropogenic N. In particular, our results suggest that 30-96% of
inputs of N from the Washington D.C. Blue Plains wastewater treatment plant were
removed *via* burial or denitrification along the Potomac river-estuarine continuum,
depending on the season (Table 1). Recent work shows that urban watersheds and river





networks can also be "transformers" of nitrogen across similar broad spatial scales, which
impacts downstream coastal water quality (Kaushal et al., 2014a). Here, we show that
the urban river-estuarine continuum also acts as a transformer and can have large impacts
on the sources, amounts, and forms of nitrogen transported to the Chesapeake Bay. Our
results showed that N transformation varied across seasons and hydrologic conditions
with important implications for anticipating changes in sources and transport of coastal
nitrogen pollution in response to future climate change. This is particularly significant,
given long-term increases in warming water temperatures of major rivers and increased
frequency and magnitude of droughts and floods in this region and elsewhere (e.g.
Kaushal et al., 2010a; Kaushal et al., 2014b).

**4.1     Spatial and Temporal Trends in N Concentrations and Loads**

The decrease in DIN concentrations with distance down-estuary is largely from

denitrification, assimilation, and burial, as indicated by the inverse relationship between
$NO_3^-$ concentrations and DOC and TON concentrations, the $NO_3^-$ isotope data, and N
mass balance data discussed below. Dilution from tidal marine waters plays a minor role
in the decrease in DIN and the incoming tidal waters may even contribute to DIN as
suggested by the decrease in DIN slope after 130 km down estuary (Boynton et al.,
1995), depending on the season. The installation of tertiary wastewater treatment
technology at Blue Plains in the year 2000 showed a significant drop in DIN
concentrations within 20-30 km of Blue Plains. However, the DIN concentrations below
30 km down-estuary were approximately the same based on an annual average, before
and after the year 2000. One explanation is that the dissolved wastewater N is



completely assimilated into particulate organic matter (supported by the inverse $NO_3^-$ *vs.*
TON or DOC relationships (Fig.s 3a and 4) or removed by denitrification (as suggested
by the isotope data discussed below) within the first 10 km down-estuary, and thus the
majority of DIN below 30 km is from other inputs than the Blue Plains wastewater
treatment plant.  For example, there are 14 other smaller wastewater treatment plants
along the Potomac River Estuary, which contribute a total of about 270 mgd (almost as
much as the amount Blue Plains contributes).  Also, our isotope mixing model data
(discussed more below) suggests nitrification (likely of upriver manure or ammonia
fertilizer inputs) and fertilizer are important sources further down-estuary; and 42% of the
land-use along the Potomac Estuary is agriculture (Karrh et al., 2007b).  A second
explanation could be related to a change in N:P ratio with distance down-estuary.
Specifically, there was a rise in estuarine salinity around 30 to 50 km down-estuary and a
coinciding increase in dissolved $PO_4^{-3}$ concentration (typical of the estuarine salinity
gradient) (Jordan et al., 2008).  When the N:P ratio fell below the Redfield Ratio of 16:1,
the estuary could shift from P limitation to N limitation (Fisher et al., 1999).  The
potential shift from P to N limitation occurred 40-50 km down-estuary, around the
estuarine turbidity maximum, which is associated with higher estuarine bacterial
productivity (Crump and Baross, 1996), and may be driving DIN removal further down-
estuary.

Mass balance indicates that TN and $NO_3^-$ loads decreased down-estuary each

season (despite inputs from the 14 other wastewater treatment plants down-estuary).  On
an annual average, it was estimated that approximately 23-27% of the loss in TN could be
attributed to denitrification, while 73-77% was lost through burial into the estuarine



sediment. This is supported by the $NO_3^-$ isotope data indicating that there was likely
denitrification (an assimilation) of $NO_3^-$, particularly within 6 km down-estuary from the
Blue Plains wastewater treatment plant (discussed further below). Over seasonal time
scales, there was a greater percent decline in TN loading during summer and fall, likely
due to warmer temperatures and increased biological transformation (attributable to high
rates of phytoplankton uptake, detrital deposition, and remineralization for subsequent
recycling) (Eyre and Ferguson, 2005; Gillooly et al., 2001; Harris and Brush, 2012;
Nowicki, 1994), which suggested that the urban river-estuarine continuum may be more
efficient at removing TN during the summer and fall. Compared to summer and fall,
winter also had a relatively high percent decline in $NO_3^-$ loads possibly driven by the
higher concentrations typically found in winter months, which could result in quicker
assimilation through first order reaction rate kinetics (Betlach and Tiedje, 1981). Since
there was no evidence for denitrification during the winter, burial could also be a
mechanism for the relative high decline in winter months, which is typical of higher
flows (Boynton et al., 1995; Milliman et al., 1985; Sanford et al., 2001). However, more
work is necessary to evaluate the fate of nitrate using ecosystem process level
measurements.

The higher total exports of TN and $NO_3^-$ to Chesapeake Bay during the winter and

spring are due to greater N inputs from the upper and lower watershed and/or greater
flow rates. The proportion of N exports attributed to Blue Plains wastewater treatment
plant were the highest in the spring, likely due to lower water residence times (Table 2),
resulting in less time for biological uptake, removal, or burial of N. The greater decline
in N loads during the spring, however, may be attributed to multiple factors, such as



greater N loads being imported from the upper estuary and higher concentrations,
compared to summer and fall (Table 1) and thus driving greater losses (from burial and
denitrification) due to first order reaction rate kinetics (Betlach and Tiedje, 1981) similar
to winter (described above), stratification that is characteristic of higher flows (Boesch et
al., 2001), and increased burial rates due to greater sediment loads during higher flows
(Milliman et al., 1985; Sanford et al., 2001). As mentioned previously, more work is
necessary regarding linking ecosystem processes and microbial dynamics with the fate of
nitrate in the estuary. Nonetheless, the decline in TN and $NO_3^-$ loads down-estuary each
season provide strong evidence for the transformation and retention of N along estuaries.

**4.2    Spatial Trends in $NO_3^-$ Sources Indicate Denitrification and Assimilation of**

**$NO_3^-$ initially Dominates and then Nitrification Dominates Further Down-**

**Estuary**

The Potomac River estuary was a transformer of wastewater N inputs from the

Washington D.C. metropolitan area to its confluence with Chesapeake Bay. The values
for $\delta^{15}N\text{-}NO_3^-$ above the wastewater treatment plant were relatively high, suggesting
upriver sources may primarily be from animal waste (Burns et al., 2009; Kaushal et al.,
2011; Kendall et al., 2007). This is consistent with a previous study, which found that
43% of N inputs to the upper Potomac River are from manure (Jaworski et al., 1992).
Effluent inputs from the Blue Plains wastewater treatment plant significantly increased
the $\delta^{15}N\text{-}NO_3^-$ values even further, yet this $NO_3^-$ signal from wastewater disappeared after
20-30 km down-estuary. The increase in $\delta^{15}N\text{-}NO_3^-$ and $\delta^{18}O\text{-}NO_3^-$ values within the first
1 to 6 km down-estuary suggest denitrification or assimilation of nitrate, due to the



lighter $\delta^{14}$N-NO$_3^-$ and $\delta^{16}$O-NO$_3^-$ isotopes being preferentially denitrified or assimilated
and leaving behind the heavier nitrate isotopes (Granger et al., 2008; Granger et al., 2004;
Kendall et al., 2007) (see further discussion below).  But the gradual decline in both
$\delta^{15}$N-NO$_3^-$ and $\delta^{18}$O-NO$_3^-$ values from 6 km to 160 km down-estuary suggests
nitrification dominates this portion of the estuary because the process of nitrification,
which converts ammonia to nitrate results in lighter nitrate isotopes being generated
through fractionation (Kendall et al., 2007; Vavilin, 2014) (see further discussion below).
However, the decline in $\delta^{15}$N-NO$_3^-$ and $\delta^{18}$O-NO$_3^-$ loads corresponding with the decline
in overall NO$_3^-$ loads down-estuary also suggests that the heavy nitrate isotopes are being
removed as well as the light isotopes.  The disappearance of $\delta^{15}$N-NO$_3^-$ and $\delta^{18}$O-NO$_3^-$
down-estuary where NO$_3^-$ concentrations are very low (~0.01 mg/l) may indicate that
assimilation or even denitrification is occurring on the remaining heavy $\delta^{15}$N-NO$_3^-$ or
$\delta^{18}$O-NO$_3^-$ after the lighter $\delta^{14}$N-NO$_3^-$ or $\delta^{16}$O-NO$_3^-$ is all used up (Fogel and Cifuentes,
1993; Vavilin et al., 2014; Waser et al., 1998a; Waser et al., 1998b).

Seasonal differences in the longitudinal trends for $\delta^{15}$N-NO$_3^-$ and $\delta^{18}$O-NO$_3^-$

suggest differences in biological transformations of nitrate due to differences in water
temperature, hydrology, and/or N inputs.  The $\delta^{15}$N-NO$_3^-$ values from effluent inputs
were likely higher in warmer months due to higher denitrification rates in the wastewater
treatment plant associated with warmer water temperatures (Dawson and Murphy, 1972;
Pfenning and McMahon, 1997), resulting in elevated $\delta^{15}$N-NO$_3^-$ values produced by
isotopic fractionation (Kendall et al., 2007; Mariotti et al., 1981).  An increase in $\delta^{15}$N-
NO$_3^-$ between 2 and 6 km down-estuary during summer and fall (Fig. 5b) further
suggested increased denitrification or biological uptake due to warmer water



temperatures and fractionation (Eyre and Ferguson, 2005; Gillooly et al., 2001; Harris
and Brush, 2012; Nowicki, 1994). The significant drop in $\delta^{15}$N-NO$_3^-$ beyond 10 km
down-estuary during summer and fall may have been due to mixing with other N sources
and increased nitrification (Wankel et al., 2006) (see further discussion below). During
the spring, there was also a significant decline in $\delta^{15}$N-NO$_3^-$ between 10 and 160 km
down-estuary, but this was likely attributed to dilution and nitrification, based on the
conservative mixing results discussed below. The lack of a significant change during the
winter, may be due to shorter residence times (Table 2) and cooler temperatures,
contributing to lower biological transformation rates. Further down-estuary, near the
mouth of the estuary, the increase in $\delta^{18}$O-NO$_3^-$ in winter and spring might indicate
denitrification in the estuary but in spring nitrate seems conservative based on the salinity
mixing plots. The decline in $\delta^{18}$O-NO$_3^-$ down-estuary in summer and fall suggest that
processes other than denitrification in the estuary are controlling the $\delta^{18}$O-NO$_3^-$, such as
nitrification.

**4.3    Isotope and Salinity Mixing Models Suggest Seasonal Patterns in N**

**Transformation Influenced by Temperature and Residence Time**

Seasonally, the ~2:1 relationship between $\delta^{15}$N-NO$_3^-$ and $\delta^{18}$O-NO$_3^-$ during

spring, summer and fall, may indicate denitrification or assimilation, but the salinity
mixing plots discussed below suggests no denitrification in the spring. The fact that the
$\delta^{15}$N:$\delta^{18}$O ratio is between 1 and 2 for summer and fall may suggests assimilation plays a
role, which is supported by previous studies which found a 1:1 relationship for
assimilation in the marine environment (Granger et al., 2004; Karsh et al., 2012; Karsh et



al., 2014).  However, other previous studies suggest that a $\delta^{15}N{:}\delta^{18}O$ ratio between 1 and
2 can also be cause by denitrifying bacteria (Granger et al., 2008; Lehmann et al., 2003).
Additionally, the divergence from the 2:1 ratio further down-estuary samples may
indicate mixing between two or more $NO_3^-$ sources, such as between atmospheric,
marine, or nitrification (Kaushal et al., 2011; Wankel et al., 2006).  Due to water column
dissolved oxygen levels averaging over 4 mg/L (data from Chesapeake Bay program, not
shown), assimilation likely dominates $NO_3^-$ removal in the water column, while
denitrification likely dominates nitrate removal from the sediment, which supported by
previous work (Cornwell et al., 2014; Kemp et al., 1990).

Denitrification is likely a sink for $NO_3^-$ during the summer and fall based on the

increases in $\delta^{15}N\text{-}NO_3^-$ and $\delta^{18}O\text{-}NO_3^-$ within 6 km down-estuary and due to warmer
water temperatures, while there is no evidence for denitrification in the winter due to
reduced biological activities typical in cooler winter temperatures (Eyre and Ferguson,
2005; Gillooly et al., 2001; Harris and Brush, 2012; Nowicki, 1994).  Nevertheless,
nitrate removal was significant in all seasons, including winter suggesting other
mechanisms, as indicated by the salinity based mixing lines.

Plots of salinity *vs.* $NO_3^-$, $\delta^{15}N\text{-}NO_3^-$, and $\delta^{18}O\text{-}NO_3^-$ were used to provide

evidence for conservative mixing, uptake, production, or contributions from other $NO_3^-$
sources.  $NO_3^-$ concentrations fell below the mixing lines during the summer, fall, and
winter, suggesting non-conservative mixing behavior due to the presence of a $NO_3^-$ sink,
such as assimilation or denitrification (Wankel et al., 2006).  During the spring $NO_3^-$
concentrations fell on the mixing line, however, suggesting that there were no important
sources or sinks.  This may be due to higher flows and shorter residence times in the



spring (Table 2), which can result in less biological transformations of $NO_3^-$. In the
salinity *vs*. $\delta^{15}N$-$NO_3^-$ and $\delta^{18}O$-$NO_3^-$ plots, when the isotope values fell below the
mixing lines, this suggested the contribution of $NO_3^-$ from sources with lower $\delta^{15}N$-$NO_3^-$
and $\delta^{18}O$-$NO_3^-$, such as fertilizer inputs or nitrification, which produces nitrate with lower
$\delta^{15}N$-$NO_3^-$ and $\delta^{18}O$-$NO_3^-$ values through fractionation (Kaushal et al., 2011; Kendall et
al., 2007). An increase in nitrification down-estuary is likely attributed to the conversion
of remineralized N to nitrate or from down-estuary inputs of wastewater ammonia that is
converted to nitrate (Middelburg and Nieuwenhuize, 2001). During the spring, $\delta^{18}O$-
$NO_3^-$, isotope values again fell mostly on the mixing line, which may indicate the
Potomac River Estuary is acting more like a transporter instead of a transformer (e.g.
Kaushal and Belt, 2012), transporting $NO_3^-$ without there being any significant sinks of
$NO_3^-$ or mixing with additional sources, likely due to lower residence times (Table 2) in
the spring. However, the fact that during the spring the $\delta^{15}N$-$NO_3^-$ values were slightly
below the mixing line indicates there may have been an increased amount of nitrate
inputs from the watershed through runoff carrying nitrate derived from nitrification.
During the winter, $\delta^{15}N$-$NO_3^-$ values also fell above the mixing line for some samples,
which suggested the contribution of heavy $\delta^{15}N$-$NO_3^-$ from an additional down-estuary
source (there are 14 other wastewater treatment plants in the lower Potomac watershed).
This was likely not the case during the summer and fall when other sources and sinks
may dominate due to greater biological activities (Eyre and Ferguson, 2005; Gillooly et
al., 2001; Harris and Brush, 2012; Nowicki, 1994) or during the spring when flows are
higher the there is more conservative behavior. Even though only surface water salinity,
nutrient, and isotope values were used in these mixing line plots, when bottom water



nutrient and isotope data was averaged with the surface water values, the mixing lines
plots and results did not change (data not shown).
**5  Conclusion**
By coupling isotope tracking techniques and a mass balance over broader spatial
and temporal scales, we found that an urban river-estuarine continuum in the Chesapeake
Bay, and likely similar estuaries globally can transform anthropogenic inputs of N over
relatively short spatial scales. Only a small fraction of N inputs from a major wastewater
treatment plant were exported out of the estuary. However, processing of N by estuaries
can vary considerably across seasons and hydrologic extremes, with greater exports
during periods of higher flows and cooler temperatures, and greater transformations and
retention during longer hydrologic residence times and warmer temperatures. In
particular, this study supports previous work, showing that non-point sources of N were
more dominate during winter and spring when runoff from the watershed and estuarine
flows were higher compared to summer and fall when the point-sources were more
dominant, due to lower flows. These differences suggest N processing in urban estuaries
would differ from those in non-urban estuaries. Also, the potential for long-term and
widespread increase in water temperatures and frequency and magnitude of droughts and
floods through climate change (Kaushal et al., 2010a; Kaushal et al., 2014b; Kaushal et
al., 2010b), will likely influence the sources and transformation of nitrogen to the
Chesapeake Bay and estuaries globally. Consequently, future efforts to manage nutrient
exports along estuaries would benefit from better understanding the interactive effects of
land use and climate variability on the sources, amounts, and transformations of N





exported to coastal waters and targeting critical times for more intensive wastewater
treatment.

**Details on Supporting Information**
• Additional site information and details on methods
• Table with site coordinates
• Table with mixing model
• Table comparing between box model (this study) and Chesapeake Bay Model.
• A figure comparing box model results with and without bottom water isotope data

**Data Availability**
Data used for the research in this paper is available through 4TU.centre at the following
DOI and URL: doi:10.4121/uuid:e68c6141-f83e-4375-ac3b-088ddf4eff51
http://doi.org/10.4121/uuid:e68c6141-f83e-4375-ac3b-088ddf4eff51

**Author contribution**
This paper is based on work from Michael Pennino's PhD dissertation. Dr. Michael
Pennino collected water samples, conducted data analysis, and wrote the manuscript. Dr.
Sujay Kaushal contributed to the study design, and provided helpful feedback on data
analysis and manuscript writing. Dr. Sudhir Murthy contributed to study design,
provided data, and contributed to manuscript revisions. Joel Blomquist contributed to
study design, sample collection, and manuscript revisions. Dr. Jeff Cornwell contributed
to manuscript revisions and provided feedback on data analysis. Dr. Lora Harris



contributed to study design, and helped with data analysis (particularly for the box model
mass balance), and manuscript writing.

**Acknowledgements**
Contact the corresponding author (michael.pennino@gmail.com) regarding the nitrate
isotope data. The historical water quality data used in this study was collected by the
Maryland Department of Natural Resources and is available free through the Chesapeake
Bay Program's Data Hub website:
(www.chesapeakebay.net/data/downloads/cbp_water_quality_database_1984_present).
This research was supported by the Washington D.C. Water and Sewer Authority. We
would like to thank Sally Bowen and Matt Hall from the Maryland Department of
Natural Resources (DNR) for their assistance in collecting monthly water samples along
the Potomac Estuary and David Brower at the U.S. Geological Survey for help in
collecting monthly river input samples for the Potomac River. We acknowledge the input
provided by Lewis Linker and Ping Wang of the US EPA Chesapeake Bay Program's
Modeling Team for providing simulated output from the CE QUAL ICEM model at three
flux boundaries in the Potomac for comparison with our box model output. Gratitude is
extended to Dr. Jeremy Testa for his suggestions regarding the box model effort. Tom
Jordan also provided helpful suggestions.





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



Table 1. Seasonal comparison of N and C inputs, exports, and losses along the Potomac River Estuary.

| | Nutrient | Total Inputs (kg/day) | % of Inputs from Blue Plains* | Net Export (kg/day) | % of Blue Plains Inputs Exported | Net Loss in Load along Estuary, Box 1 to 6 (kg/day) | % Net Loss in Load along Estuary, Box 1 to 6 | Net Loss in Load along Estuary, Box 1 to 5 (kg/day) | % Net Loss in Load along Estuary, Box 1 to 5 | Net Loads from Bay to Estuary (kg/day) |
|---|---|---|---|---|---|---|---|---|---|---|
| Winter | TN | 49150 ± 30323 | 10 ± 13 | 19844 ± 13728 | 3.7 ± NA | 27369 ± 14597 | 54 ± 40 | 16426 ± 9509 | 28 ± 25 | 473 ± 414 |
| Spring | TN | 135317 ± 14614 | 8 ± 0.8 | 68431 ± 48060 | 71 ± 20 | 49672 ± 52116 | 36 ± 43 | 29515 ± 32908 | 26 ± 21 | -127 ± 480 |
| Summer | TN | 13888 ± 596 | 38 ± 3 | 4853 ± 8326 | 19 ± 11 | 7155 ± 8370 | 75 ± 75 | 5739 ± 1832 | 44 ± 21 | 380 ± 164 |
| Fall | TN | 15334 ± 3700 | 47 ± 13 | -1613 ± 12124 | 18 ± 10 | 15364 ± 12548 | 112 ± 95 | 4140 ± 6607 | 30 ± 43 | 264 ± 290 |
| Winter | NO$_3^-$ | 37749 ± 23574 | 5.7 ± 4.6 | 2080 ± 6235 | 3 ± NA | 31791 ± 7417 | 93 ± 29 | 26299 ± 10069 | 74 ± 33 | 32 ± 58 |
| Spring | NO$_3^-$ | 95395 ± 10416 | 7.4 ± 0.6 | 30039 ± 161747 | 52 ± 70 | 40206 ± 161977 | 60 ± 187 | 30998 ± 26791 | 46 ± 34 | 8 ± 109 |
| Summer | NO$_3^-$ | 7066 ± 364 | 49 ± 6.3 | 105 ± 4130 | 17 ± 2 | 5166 ± 4143 | 96 ± 141 | 4223 ± 763 | 77 ± 19 | 11 ± 10 |
| Fall | NO$_3^-$ | 10526 ± 3006 | 53 ± 18.2 | -204 ± 6278 | 13 ± 35 | 7291 ± 6812 | 108 ± 181 | 5637 ± 6817 | 85 ± 122 | 13 ± 35 |
| Winter | δ$^{15}$N-NO$_3^-$ | 130 ± 10 | 4 ± 0.4 | 4 ± NA | 2.7 ± NA | 130 ± NA | 97 ± NA | 77 ± NA | 68 ± NA | 86 ± NA |
| Spring | δ$^{15}$N-NO$_3^-$ | 374 ± 3 | 7 ± 0.1 | 170 ± 547 | 52 ± 136 | 88 ± 547 | 48 ± 136 | 42 ± 71 | 26 ± 31 | -412 ± 1471 |
| Summer | δ$^{15}$N-NO$_3^-$ | 30 ± 1 | 53 ± 1.6 | 5 ± 1 | 17 ± 3 | 27 ± 1 | 83 ± 3 | 18 ± 1 | 83 ± 3 | NA |
| Fall | δ$^{15}$N-NO$_3^-$ | 40 ± 5 | 55 ± 5.8 | 7 ± 8 | 13 ± 68 | 26 ± 8 | 87 ± 105 | 26 ± 13 | 87 ± 105 | NA |

TN = Total Nitrogen. NA – indicates there was only one month with data for that season and thus no S.E. value.
*Blue Plains is a wastewater treatment plant.





Table 2. Comparison of mean seasonal discharge and residence time within the Potomac
River Estuary

|  | Mean Discharge ($m^3/s$) | Mean Residence time (days) |
|---|---|---|
| Winter | $187 \pm 60$ | $26 \pm 18$ |
| Spring | $545 \pm 214$ | $57 \pm 36$ |
| Summer | $81 \pm 29$ | $129 \pm 85$ |
| Fall | $81 \pm 27$ | $196 \pm 102$ |

Data is based on discharge and box model results for the period from April 2010 to
March 2011.






















Figures
Figure 1. Map showing the Potomac River sampling stations (black diamond) and the
location of the Blue Plains Wastewater Treatment plant (WWTP, black X) just south of
Washington D.C., within the Chesapeake Bay watershed.  The larger figure shows the
location of monthly extensive synoptic surveys sites and the smaller figure on upper left
shows the locations of the shorter intensive synoptic surveys.  The larger figure also
shows the location for the historical Maryland DNR surface water sampling sites.
Figure 2. Plot of the Potomac Estuary depth with distance down-estuary showing the
location of the 6 boxes used in the box model calculations.
Figure 3. Longitudinal patterns in Potomac River Estuary: (a) mean annual dissolved
inorganic nitrogen (DIN) and total organic nitrogen (TON) spanning 1997 to 2005, (b)
mean seasonal DIN before year 2000 (1994 to 1999), and post 2000 (2001 to 2012), and
(c) mean (1994 to 2012) seasonal molar N:P ratio (DIN/$PO_4^{-3}$), with salinity averaged
from all seasons (1984 to 2008).  Note: errors bars are provided, but S.E. is relatively
small compared to concentrations.  This data was obtained from the Maryland DNR and
the Chesapeake Bay Program Data Hub.
Figure 4. Comparison of $NO_3^-$ $vs$. dissolved organic carbon (DOC). N and C data was
obtained from the Maryland DNR and the Chesapeake Bay Program Data Hub for this
study period.
Figure 5. Trends in (a) $\delta^{15}N$-$NO_3^-$, (b) $\delta^{18}O$-$NO_3^-$, and (c) percent contribution of nitrate
from wastewater, the atmospheric, and nitrification, based on isotope mixing model, with
distance down-estuary from wastewater treatment plant input.  Error bars are standard
errors of the mean.  N = 1 for winter, N = 3 for spring and fall, and N = 2 for summer.
Figure 6. (a) Plot of $\delta^{15}N$-$NO_3^-$ $vs$. $\delta^{18}O$-$NO_3^-$ of nitrate from effluent water samples and
Potomac River Estuary samples, showing samples from different locations along the
estuary; the grey arrow indicates the 2:1 relationship characteristic for denitrification; and
(b) Same plot as (a), but seasonally and without the effluent or wastewater outfall values.
Not included in these plots is the box indicating the region where atmospheric nitrate
samples generally lie, from -10 to +15 for $\delta^{15}N$-$NO_3^-$ and from 60 to 100 for $\delta^{18}O$-$NO_3^-$.
Figure 7. Comparison of salinity $vs$. (a) $NO_3^-$, (b) $\delta^{15}N$-$NO_3^-$ and (c) $\delta^{18}O$-$NO_3^-$.  Mixing
lines connect the mean $NO_3^-$ concentration or isotope values at the lowest and highest
salinity values.  Error bars are standard errors of the mean.  For panel (a), N = 3 for all
seasons, for panels (b) and (c), N = 1 for winter, N = 3 for spring and fall, and N = 2 for
summer.  Mixing line equations for $NO_3^-$ concentrations and isotopes were obtained from
Middelburg and Nieuwenhuize (2001).  $NO_3^-$ data was obtained from the Maryland DNR
and the Chesapeake Bay Program Data Hub, covering spring 2010 to spring 2011, the
same dates as the $NO_3^-$ isotope data.



Figure 8. Comparing the TN fluxes along the Potomac River Estuary estimated from the
Box Model used in this study and from the results from the Chesapeake Bay nutrient
model.

Figure 9. Correlation between the fluxes estimated from the Box Model used in this study
and the Chesapeake Bay nutrient model.

Figure 10. Seasonal Box Model results showing how (a) TN, (b) $NO_3^-$, and (c) $\delta^{15}N$-$NO_3^-$
loads vary down-estuary. Error bars are standard errors of the mean. For panels (a) and
(b), N = 3 for all seasons. For panel (c), N = 1 for winter, N = 3 for spring and fall, and N
= 2 for summer. TN and $NO_3^-$ data was obtained from the Maryland DNR and the
Chesapeake Bay Program Data Hub.






Figure 1.

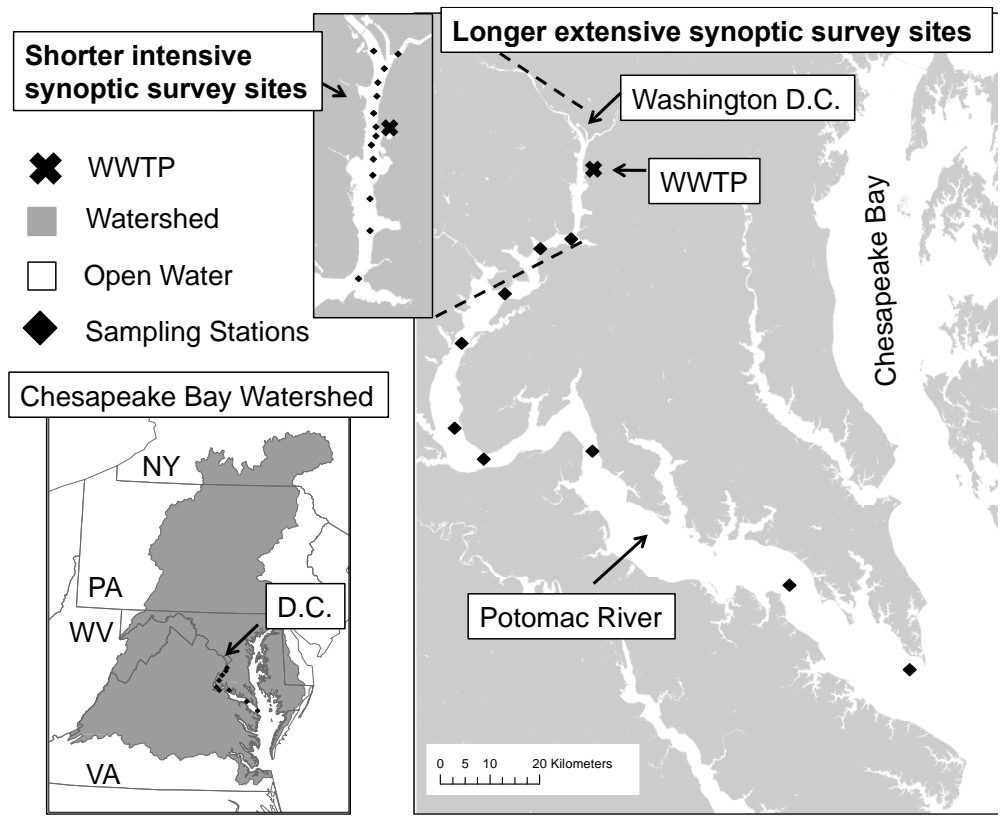













Figure 2.

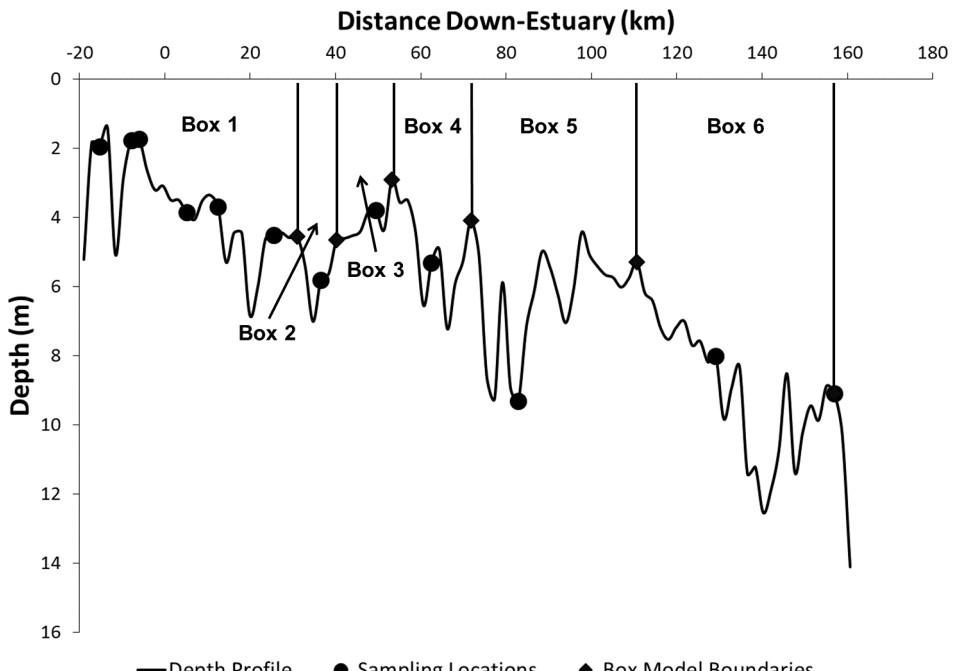






Figure 3.

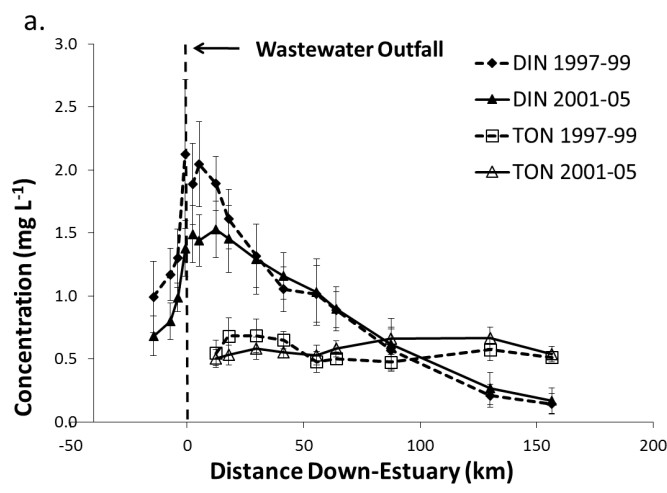

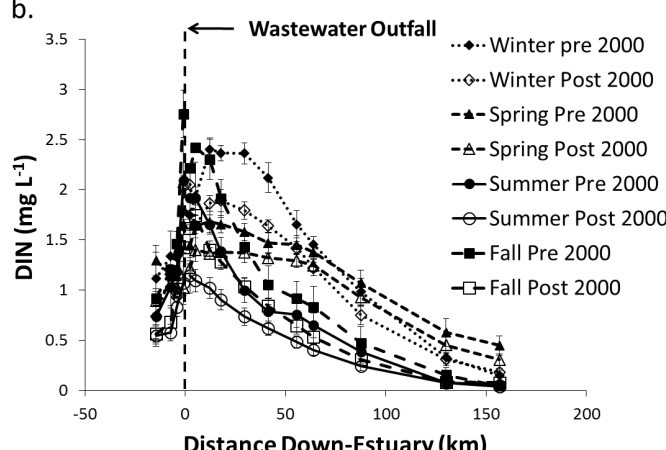

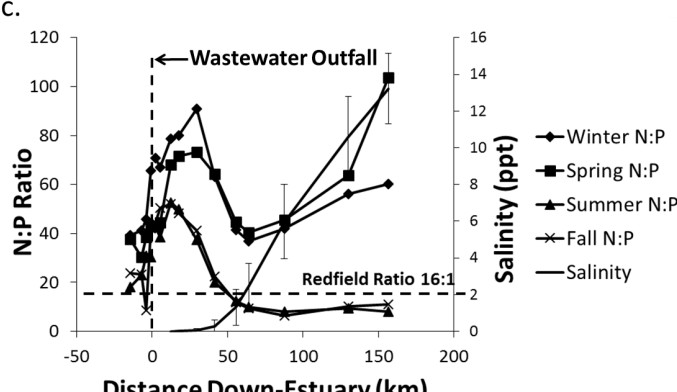




Figure 4.

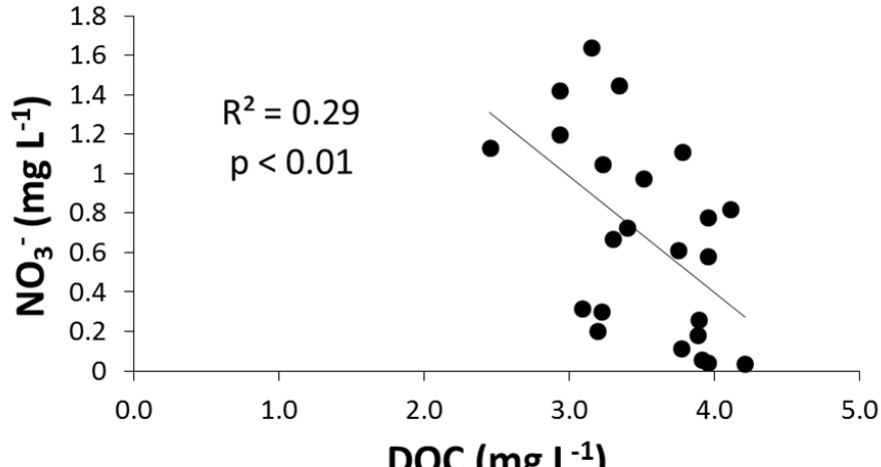




Figure 5.

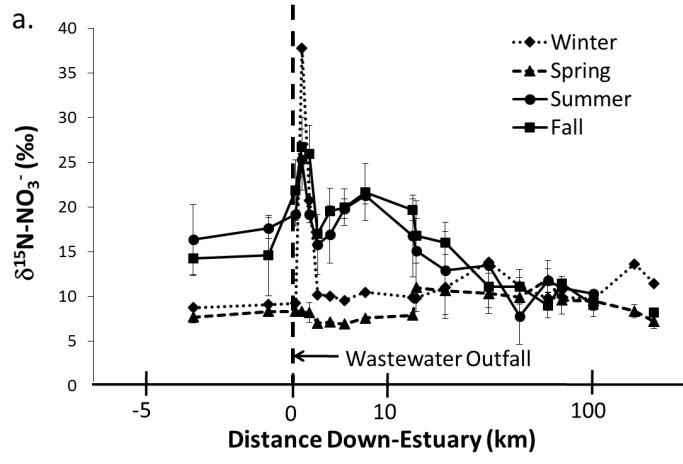

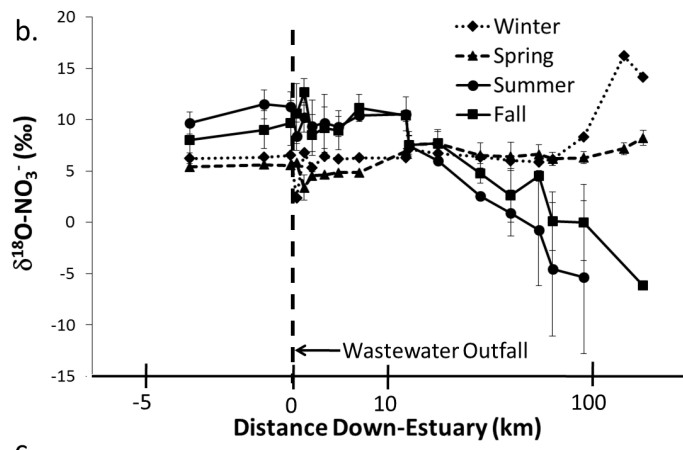

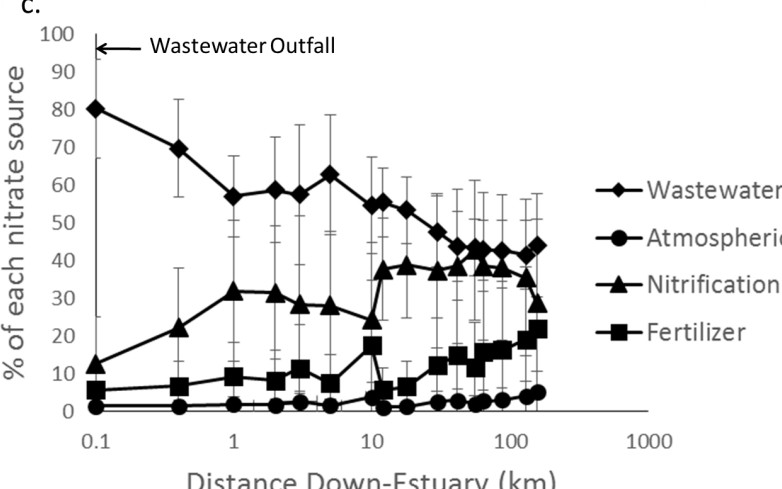





Figure 6.

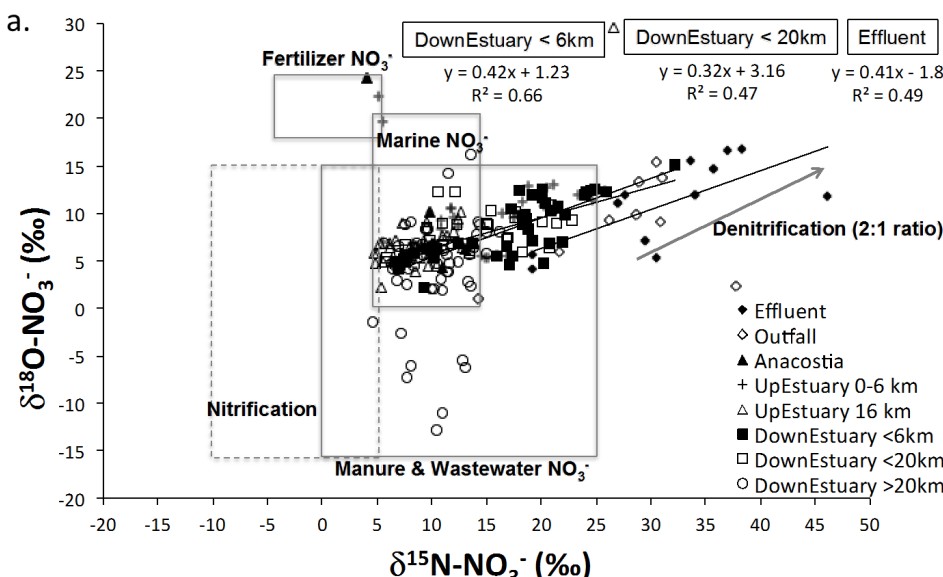

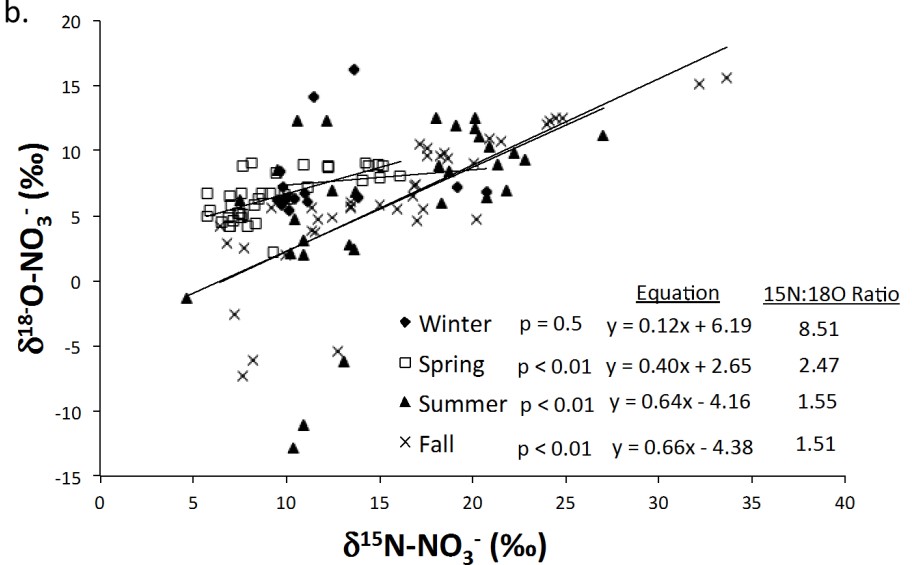




Figure 7.

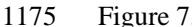

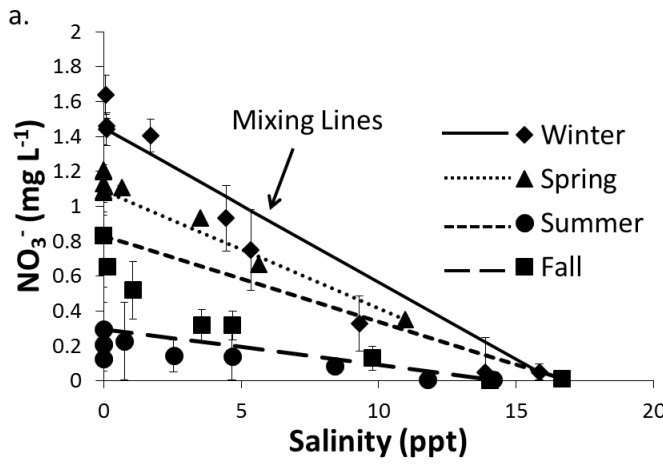

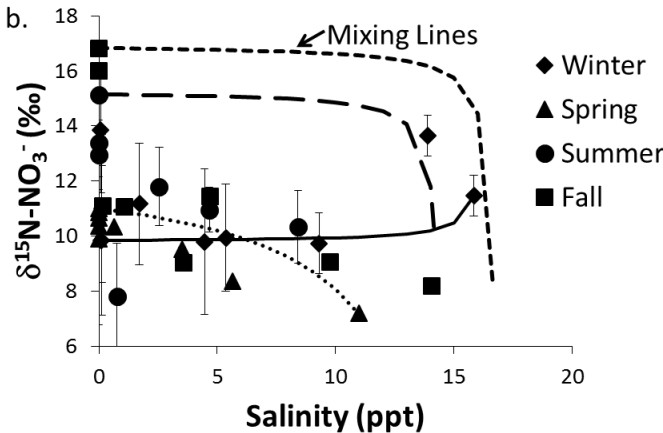

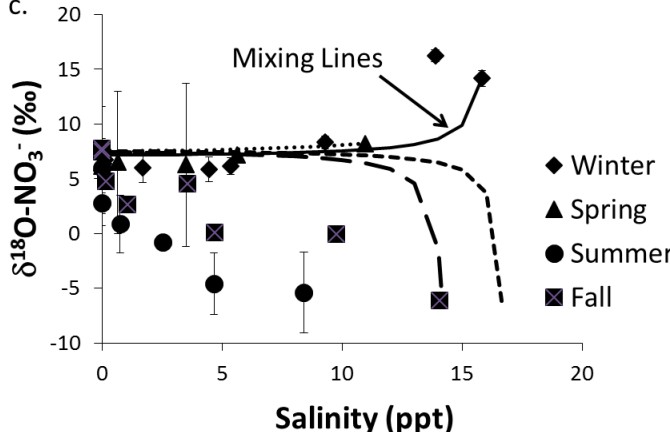






Figure 8.

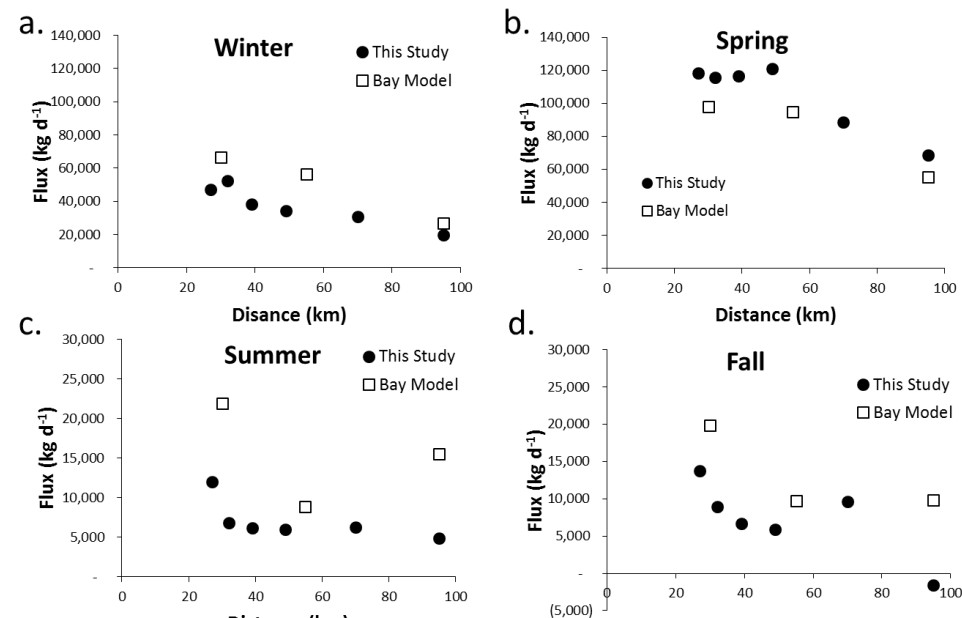




Figure 9.

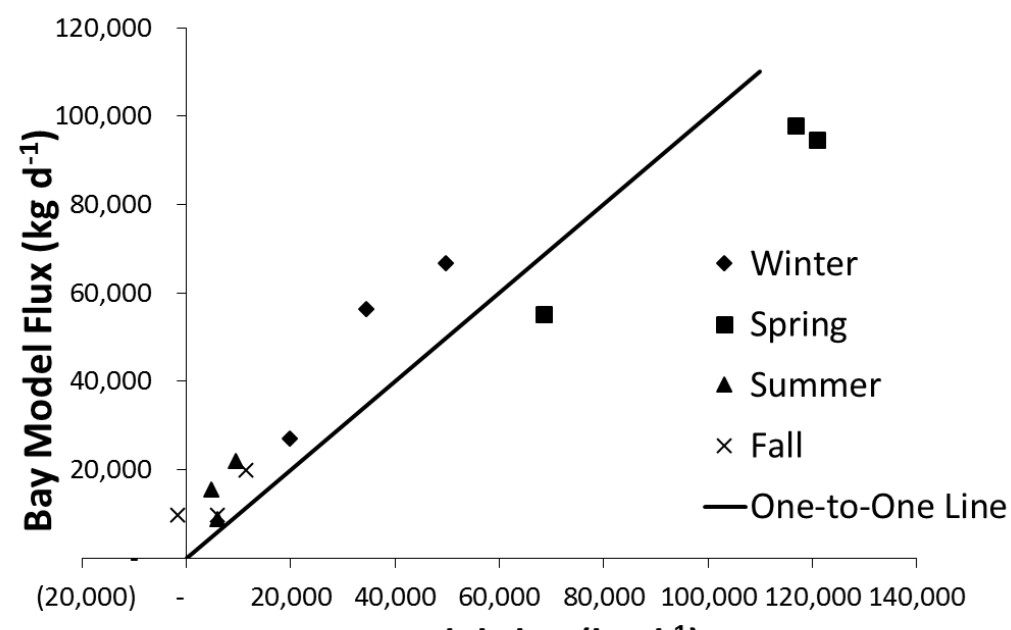





Figure 10.

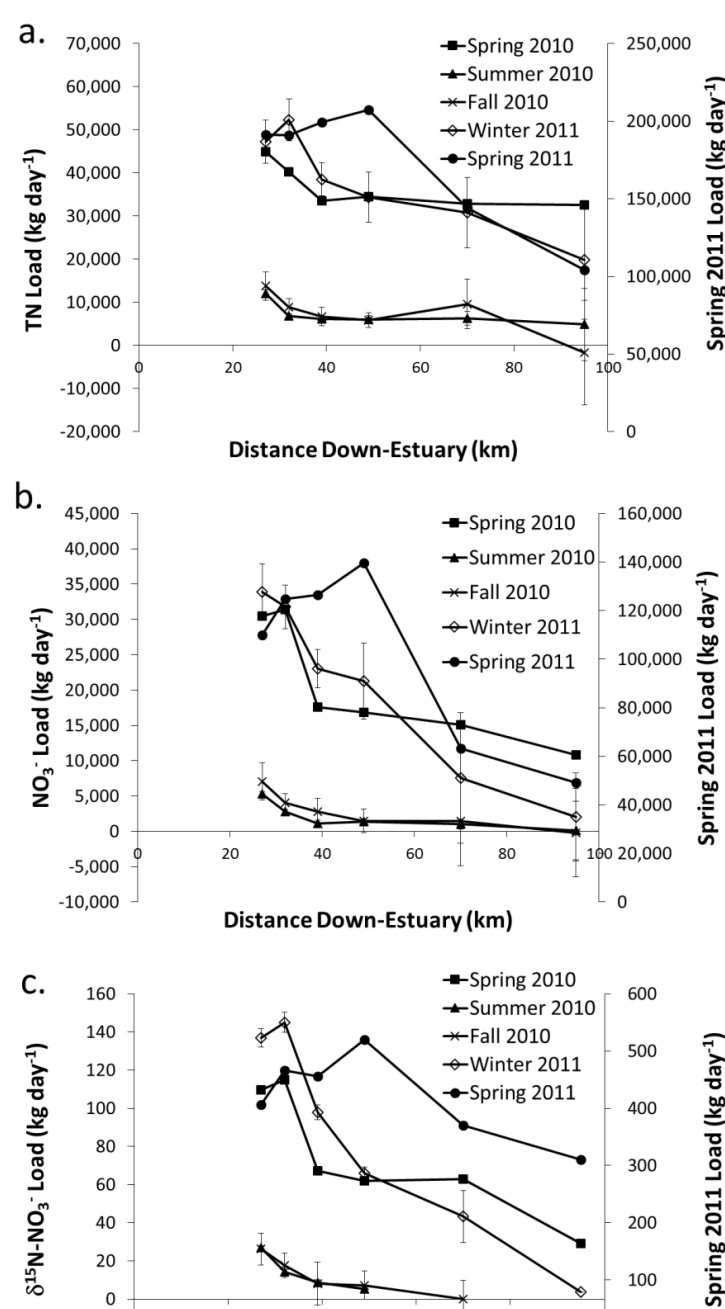
