# Peer review of "Sources and Transformations of Anthropogenic Nitrogen along an Urban River- Estuarine Continuum"

_Biogeosciences, 2016_

## Referee Comment (RC1) · A.F. Bouwman (Referee) · 4 Aug 2016

bg-2016-264 Title: Sources and Transformations of Anthropogenic Nitrogen along an Urban River-Estuarine Continuum Authers: M.J. Pennino et al.

This is an interesting paper that answers a number of important research questions, covering the attribution of sources, transformations of nitrogen, and the impact of the hydrological conditions over the Potomac river-estuary continuum of 150 km. Isotope and mass balance approaches are combined to track nitrogen sources and transformations along this distance. The results of this work can be very helpful in designing strategies to manage the water quality of this densely populated river basin.

The paper is well structured, and reads easily. I have a few concerns and a number of minor comments and suggestions.

Lines 290-304: the reasoning why the 14 down-stream WWTPs have little effect is completely unclear to me. Particularly 301-304 is not clear.

With all the uncertainties associated with the mixing model (line 204-206, line 214-216) and the caution (use for illustrative purposes only), I wonder if it makes sense to present it at all, since I do not know what the meaning is of "illustrative purposes" is if I do not know the uncertainty. The attribution of sources in the text looks pretty certain (no word about the illustrative purpose), and the uncertainty ranges are very small. That is surprising to me, and I wonder how these ranges are obtained? Is it the same error propagation method discussed in lines 267-274?

The range for the contribution of denitrification to the TN decline of 23-27% (Line 478; Line 543) suggests it is an uncertainty, but is simply is two different estimates, a direct and indirect one. It is possible to provide a real uncertainty here? The Burial rate presented by Boynton et al. is an average for upper and lower Potomac estuary, and it is not clear if this calculation was done by the original paper or in this study, but it is probably quite and uncertain number. Similar question about the average denitrification rate.

In various places the authors indicate that a statement or reason for a phenomenon is described "below": e.g. line 292, 311, 318, 434, 514, 529, 547,597, 633. For readers this is awkward, because they start looking where this could be, because they want the explanation for something they read. Now either the word below can be avoided by placing the discussion referred to directly after the statement, or the explanation comes first, and then the concluding statement.

Minor comments

-I am not sure if present and past tense is consistently used correctly in the results and

discussion sections. Please check.

-I do not know how many times the words "additionally", "suggest" and "suggesting" are used, but it is a lot. Please try to vary.

-Line 126 and line 134: confusion between current concentration of 2.3 mg/L and 2001-2008 concentration of 4.1 mg/L. What is current? Has it gone down further, or what is the reason of the difference?

-Line 187: atmospheric deposition.

-It is not clear to me if the sampling locations in Figure 2 correspond to those in Figure 1. For example, the first point at about -17 km is not in Figure 1.

-Line 232: insert that after indicate.

-What is at distance zero in Figure 2? Is that the WWTP?

-Lines 267-274: I assume that the errors are expressed as standard deviations? If so, please mention.

-Line 295-296: The isotope signal for the Blue Plains has been mentioned previously. These references provide numbers for the 14 down-stream WWTPs, I assume, but it suggests that they are for Blue Plains.

-Line 362: what is directly down-estuary?

-Line 385: up-estuary? Line 386: 2km down-estuary? Is it possible to attach a code to sampling locations, show that in Figure 1 (and 2) and refer to those codes instead of these up and down indicators of locations?

-Line 506: it is not clear if there is a long-term warming or increasing warming; are temperatures warming?

-Line 528: What is the unit "mgd".

-Line 546: is it AND assimilation?

-Lines 565 and 671: shorter times instead of lower.

-Lines 547-551: remineralization leads to addition of TN, so I'd attribute a decrease in TN to uptake and subsequent deposition.

-The header 4.2 and 4.3 read like a conclusion, not a section header. In addition, it looks like in 4.2 a few words are missing (indicate that) and dominate (two processes dominate); If this is actually the intention, then please be consistent, and change 4.1 in a similar way.

-Line 634: may suggest→suggest or indicate? Otherwise 2x suggest

-Line 638: caused.

-Line 644: is supported.

-Line 674: nitrate produced by nitrification.

-Line 681: delete "the" and change the order of the sentence: there is more conservative behavior when flows are larger.

-Line 695: dominant.

Lex Bouwman

---

## Referee Comment (RC2) · A. E. Giblin (Referee) · 14 Aug 2016

General comments: The authors have used mass balance models, constrained with stable isotopes to identify the sources and fate of nitrogen in the Potomac river-estuary continuum. A large outfall from a tertiary sewage treatment plant contributes 8-47% of the total upstream N loading depending upon the season. The goal of the study is to evaluate how well this high N load is attenuated before being transported downstream to Chesapeake Bay. They highlight the importance of making these measurements under different flow regimes since many studies have shown that N assimilation can be very sensitive to discharge. The approach the authors have taken serves as an excellent model for other studies but also illustrates some of the difficulties in this approach. Overall it is a very useful contribution and the findings can also help inform management to help understand where source reduction might be the most effective. I do believe the authors could make better use of the data they have to constrain some of the possible findings. Specific comments Line 141 and 381 – this spans the period of time both before and after the nitrate in the effluent decreased by nearly half from the treatment plant. Would Fig 4 be a better fit if the data was separated into pre and post periods? 173-179 – Given how well studied this system is I suspect there is nitrite data? Based upon that data is nitrite high enough to be of concern? 192 – single values are given here but fig. 6 shows a range of values which makes more sense. These uncertainties could be incorporated into the estimates. 195, 202 I was a bit confused by this, aren't manure based fertilizers also used in region as well? The discussion is section 4.2 suggests this is a major input. Was the nitrate fertilizer value chosen because the authors know that is what is used here and manure is only important upstream? 204 – I thought putting this in the methods was an odd way to present this. In spite of the uncertainty the isotopes do put some constraints on the data. I think it makes more sense to present the data and then discuss the limitations and errors. 218, and section starting on 422 – How are additional lateral inputs of freshwater being dealt with here? There has been a lot of modeling of this region so I'm sure they are known but it would be good to state the assumptions/data behind this. Lateral sources of freshwater might also have significant nitrate concentrations and different isotopic signatures. If the amounts are trivial this should be stated. 290 – I did not like the assumption that these other treatment plants would have little impact. An additional 32% is significant and depending upon where it is added could be very important. The locations of the plants are not given but could this account for the lack of change further downsteam post treatment change (line 367)? The authors also don't mention what types of plant these other WTP are (secondary or tertiary). Some secondary plants get to very high values 15N values if there is extensive open aeration. I agree that the net impact of all of these plants will probably be to underestimate biological assimilation but it would seem to be beneficial to constrain the system to the extent possible.

Instead, it is dismissed here and then brought up in the discussion (526) where we find out that the total flows are nearly as large as the Blue Plains plant. It is then brought up again on line 674-5. I believe the authors will have a more robust story if these plants are incorporated into the model. 414 – Fractionation will only be apparent if only part of the pool is used. While this would seem to be the case, because, nitrate does not completely disappear, there is data from a variety of sources that indicates that sometimes denitrification occurs in hot spots (like hyporheic zones) where part of the pool is completely denitrified without any change in the isotopic composition. I think this at least deserves some mention. 458-475 - I think this could be made clearer. I was initially quite concerned about the very large error bars. The authors attribute some of this to the uncertainties in the last box but in looking at Table 1 things don't improve that much when box 6 is omitted. If I assume all of the seasons are of equal length (3 months) than the seasonal averages presented on 458-460 work out to a loss of 9.03x106 kg/year. With the propagated error this is nearly +/- 100%! But this can be compared to the independent estimates of burial and denitrification rates presented in Boynton et al. 1995 (lines 469-474) of 9.89 x106 kg/year. This agreement is quite good, and I wonder if these huge error bars are due to the method of error propagation. A monte carlo approach might result in smaller errors. I think I would point out the good agreement before going on to attribute the % loss to burial and denitrification.

Section 4.2 and 4.3 This discussion is quite long and discusses many possible explanations for some of the data but seldom comes to strong conclusions. The authors have some great data here, I'm not sure they are making the most of it. These section contains a lot of statements such as those on line 607 "15N-NO3 values were likely higher in warmer months due to denitrification" since monthly measurements were made don't you know whether or not this is true? No mention is made on line 198 of seasonal changes so I had assumed this was not true. If it is true, the model should be run with different values for different seasons correct? The isotopes are not sufficient to tell when nitrate removal is due to assimilation or denitrification but doesn't the Boynton et al. 1995 data provide some insights that could be used? As mentioned above, on

lines 674 the possible role of N from additional treatment plants is brought up when it had been dismissed earlier. So, overall, I think the authors may be able to constrain this system better and come up with more robust conclusions.

Conclusions – the importance of hydrology and temperature in N transformation is a critical issue for management and often discussed but removal is also a function of total load. I agree with all of the authors statements but differences in the N behavior in the manuscript is largely discussed by season and I think the conclusions could do a better job talking about all three factors.

---

## Author Comment (AC1) · 7 Oct 2016

Biogeosciences Discussion Paper Response to Referee Comments

Michael J. Pennino et al. "Sources andTransformations of Anthropogenic Nitrogen along an Urban River-Estuarine Continuum" doi:10.5194/bg-2016-264

Note: We copied the referee comments below and responded directly after each comment or question. The referee comments have a hyphen at the beginning of each comment. Our responses follow directly after each referee comment or question.

Response to Referee 1 This is an interesting paper that answers a number of important research questions, covering the attribution of sources, transformations of nitrogen,

and the impact of the hydrological conditions over the Potomac river-estuary continuum of 150 km. Isotope and mass balance approaches are combined to track nitrogen sources and transformations along this distance. The results of this work can be very helpful in designing strategies to manage the water quality of this densely populated river basin.

The paper is well structured, and reads easily. I have a few concerns and a number of minor comments and suggestions.

-Lines 290-304: the reasoning why the 14 down-stream WWTPs have little effect is completely unclear to me. Particularly 301-304 is not clear.

We have modified this paragraph and no longer say that we assume the 14 WWTPs have little effect, but focus instead on how their effect is only to increase the loads along the estuary, and thus counteract the overall decline in loads that are observed along the estuary. And we also emphasize how there is likely little impact on the isotope levels due to the average isotope levels from primary and secondary WWTPs being much lower than what was measured at the Blue Plains WWTP (see further details in the response below).

-With all the uncertainties associated with the mixing model (line 204-206, line 214-216) and the caution (use for illustrative purposes only), I wonder if it makes sense to present it at all, since I do not know what the meaning is of "illustrative purposes" is if I do not know the uncertainty. The attribution of sources in the text looks pretty certain (no word about the illustrative purpose), and the uncertainty ranges are very small. That is surprising to me, and I wonder how these ranges are obtained? Is it the same error propagation method discussed in lines 267-274?

The uncertainties in the nitrate isotope sources came from the literature, except for wastewater nitrate which came from averaging about a year of monthly samples.

The method of error propagation described in lines 267-274 was only used for the box

model mass balance estimates - not the isotope mixing model, which used a Bayesian approach (described in the methods).

We believe the results of the isotope mixing model are still useful (such as to show trends over distance) despite the potential variability. Also, the other reviewer liked the mixing model approach for illustrative purposes and thought it could be used to make stronger conclusions. For example, seasonal endmembers could provide more confidence in the results because we found that seasonality/temperature mattered in endmembers. Many isotopic studies do not always take this into account and sometimes they just use literature values – our work showed that there are important seasonal variations and thus seasonal changes in the other endmembers may need to be captured.

We have updated the discussion section to discuss how the longitudinal trends in nitrate sources along the Potomac Estuary correspond with the other results of this study and how future use of the isotope mixing model would benefit from conducting the model separately for each season to better constrain the differences between seasons.

-The range for the contribution of denitrification to the TN decline of 23-27% (Line 478; Line 543) suggests it is an uncertainty, but is simply is two different estimates, a direct and indirect one. It is possible to provide a real uncertainty here? The Burial rate presented by Boynton et al. is an average for upper and lower Potomac estuary, and it is not clear if this calculation was done by the original paper or in this study, but it is probably quite and uncertain number. Similar question about the average denitrification rate.

The estimates of % burial and % denitrification have been modified. They still only use information from the Boynton et al. (1995) paper (Table 6). This paper did not provide uncertainties in their estimates. However, recent measurements of burial (manuscript in preparation) and denitrification (Cornwell et al. 2016) are in line with these estimates. And we will be able to use the information from the manuscript in preparation, which is

providing a budget for the Potomac Estuary with detailed flux estimates that will help constrain the estimates of the proportion of burial, denitrification, and assimilation for this paper.

Citation: Cornwell et al. 2016. Sediment-Water Nitrogen Exchange along the Potomac River Estuarine Salinity Gradient. JOURNAL OF COASTAL RESEARCH 32:776-787

-In various places the authors indicate that a statement or reason for a phenomenon is described "below": e.g. line 292, 311, 318, 434, 514, 529, 547,597, 633. For readers this is awkward, because they start looking where this could be, because they want the explanation for something they read. Now either the word below can be avoided by placing the discussion referred to directly after the statement, or the explanation comes first, and then the concluding statement.

We have removed most of the instances where we say "discussed below" because it was either unnecessary or sufficiently discussed in that section.

Minor comments -I am not sure if present and past tense is consistently used correctly in the results and discussion sections. Please check.

We have checked this and fixed any incorrect use of past or present tense.

-I do not know how many times the words "additionally", "suggest" and "suggesting" are used, but it is a lot. Please try to vary.

We have removed or changed several of the instances where we previously used "additionally", "suggest" and "suggesting". Other words we used were "indicate" or "show" instead of "suggest," for example.

-Line 126 and line 134: confusion between current concentration of 2.3 mg/L and 2001-2008 concentration of 4.1 mg/L. What is current? Has it gone down further, or what is the reason of the difference?

To clarify I added that the 2.3 mg/L value was from 2009 and the 4.1 mg/L was from

directly after the year 2000.

-Line 187: atmospheric deposition.

We added the word deposition after atmospheric.

-It is not clear to me if the sampling locations in Figure 2 correspond to those in Figure 1. For example, the first point at about -17 km is not in Figure 1.

We have updated Figure 2 so that it lines up with Figure 1, such that zero is at the location of the WWTP. And the first sample point is located at -6 km (or 6 km up-estuary from the WWTP)

-Line 232: insert that after indicate.

We have added the word "that" here

-What is at distance zero in Figure 2? Is that the WWTP?

We have updated Figure 2 so that it lines up with Figure 1, such that zero is at the location of the WWTP.

-Lines 267-274: I assume that the errors are expressed as standard deviations? If so, please mention.

We clarified that error propagation was done using standard errors.

-Line 295-296: The isotope signal for the Blue Plains has been mentioned previously. These references provide numbers for the 14 down-stream WWTPs, I assume, but it suggests that they are for Blue Plains.

We've added that these values were for typical WWTP nitrate isotopes to make sure they were not confused with Blue Plains values.

-Line 362: what is directly down-estuary?

We added to this sentence that the Blue Plains WWTP is directly down-estuary.

[Figure]

-Line 385: up-estuary? Line 386: 2km down-estuary? Is it possible to attach a code to sampling locations, show that in Figure 1 (and 2) and refer to those codes instead of these up and down indicators of locations?

We believe that updating Figure 2 so that both show the Blue Plains WWTP at distance zero will help. We do not think attaching codes is necessary. For clarification we also added to the text and figure captions that the distance up-estuary or down-estuary was in references to distance from the Blue Plains WWTP.

-Line 506: it is not clear if there is a long-term warming or increasing warming; are temperatures warming?

We removed the word "warming" before "water temperatures," for clarification.

-Line 528: What is the unit "mgd".

This unit is defined earlier in the manuscript (line 128).

-Line 546: is it AND assimilation?

Yes we made the appropriate revision.

-Lines 565 and 671: shorter times instead of lower.

We changed lower to shorter here.

-Lines 547-551: remineralization leads to addition of TN, so I'd attribute a decrease in TN to uptake and subsequent deposition.

We changed this so that it just says "attributable to high rates of phytoplankton uptake and detrital deposition."

-The header 4.2 and 4.3 read like a conclusion, not a section header. In addition, it looks like in 4.2 a few words are missing (indicate that) and dominate (two processes dominate); If this is actually the intention, then please be consistent, and change 4.1 in a similar way.

We have changed the headers of 4.2 and 4.3 to no longer read as conclusions. They now read as: "4.2 Spatial Trends in NO3- Sources and Role of Denitrification, Assimilation and Nitrification" and "4.3 Isotope and Salinity Mixing Models and Influence of Temperature and Residence Time."

-Line 634: may suggest!suggest or indicate? Otherwise 2x suggest

We changed may suggests to may suggest.

-Line 638: caused.

We changed "cause" to "caused."

-Line 644: is supported.

We added the word is.

-Line 674: nitrate produced by nitrification.

We changed "derived from" to "produced by."

-Line 681: delete "the" and change the order of the sentence: there is more conservative behavior when flows are larger.

We have incorporated these changes.

-Line 695: dominant.

We have changed the word "dominate" to "dominant."

Response to Referee 2

General comments: -The authors have used mass balance models, constrained with stable isotopes to identify the sources and fate of nitrogen in the Potomac river-estuary continuum. A large outfall from a tertiary sewage treatment plant contributes 8-47% of the total upstream N loading depending upon the season. The goal of the study is to evaluate how well this high N load is attenuated before being transported downstream

to Chesapeake Bay. They highlight the importance of making these measurements under different flow regimes since many studies have shown that N assimilation can be very sensitive to discharge. The approach the authors have taken serves as an excellent model for other studies but also illustrates some of the difficulties in this approach. Overall it is a very useful contribution and the findings can also help inform management to help understand where source reduction might be the most effective.

-I do believe the authors could make better use of the data they have to constrain some of the possible findings.

We have incorporated the reviewer's suggestions to help make better use of the data.

Specific comments -Line 141 and 381 – this spans the period of time both before and after the nitrate in the effluent decreased by nearly half from the treatment plant. Would Fig 4 be a better fit if the data was separated into pre and post periods?

The Figure 4 caption already said it was "for this study period," but I added "(2010-2012)" for clarification.

-173-179 – Given how well studied this system is I suspect there is nitrite data? Based upon that data is nitrite high enough to be of concern?

For Potomac Estuary stations TF2.1 through LE2.3 (stations from the top of the estuary to the bottom of the estuary) the mean nitrite concentration from 2010-2012 is 0.013 mg/L and the minimum = 0.0055 mg/L and maximum = 0.0183 mg/L. The mean nitrite is about 2.4% of the mean nitrate+nitrite concentration.

Fawcett et al. 2015 says "If nitrite is present in seawater, even at levels <= 0.5% of nitrate+nitrite, it can noticeably affect the measured $\delta$18O of nitrate+nitrite (Casciotti and McIlvin, 2007; Granger and Sigman, 2009; Granger et al., 2006). This is because, during bacterial conversion to N2O, nitrite is subject to a smaller fractional loss of O atoms than nitrate (3/4 versus 5/6) such that O isotopic fractionation during nitrite reduction to N2O is lower (by $\sim$25‰ than that for N2O generated from nitrate with the

same initial $\delta$18O(Casciotti et al., 2007)."

Consequently, even though the nitrite levels are a very small portion of the nitrate+nitrite, it is possible there is an impact. We added text to the methods to further acknowledge this and included the proportion of nitrite in the samples for the reader. We have also contacted UC Davis Stable Isotope Facility, who processed our water samples for nitrate isotopes and we will see if there is a way to correct for the effects of nitrate after the fact, if we now incorporate nitrite concentrations.

-192 – single values are given here but fig. 6 shows a range of values which makes more sense. These uncertainties could be incorporated into the estimates.

The nitrate source end member values from the literature has standard deviations associated with them (provides in lines 194-198) and these uncertainties were already incorporated into the nitrate isotope mixing model estimates.

-195, 202 I was a bit confused by this, aren't manure based fertilizers also used in region as well? The discussion is section 4.2 suggests this is a major input. Was the nitrate fertilizer value chosen because the authors know that is what is used here and manure is only important upstream?

There are 171 confined animal feeding operations (CAFOs) in Upper Potomac, above DC and there are 25 CAFOs in the lower Potomac below DC. This information was added to this section of the manuscript with a citation where the CAFO data were obtained.

-204 – I thought putting this in the methods was an odd way to present this. In spite of the uncertainty the isotopes do put some constraints on the data. I think it makes more sense to present the data and then discuss the limitations and errors.

This is a good suggestion, but we respectfully disagree with moving this section to the discussion. We feel that it should be presented before the results so that the reader knows the potential limitations of the isotope mixing model prior to reading the

results. We are appreciative of the experience this reviewer has with these data, but we have encountered concerns from other reviews regarding these results and are being responsive to those concerns in this organizational structure.

-218, and section starting on 422 – How are additional lateral inputs of freshwater being dealt with here? There has been a lot of modeling of this region so I'm sure they are known but it would be good to state the assumptions/data behind this. Lateral sources of freshwater might also have significant nitrate concentrations and different isotopic signatures. If the amounts are trivial this should be stated.

We assume that the lateral inputs will show up in the samples that are from the main stem of the Potomac River Estuary and invoke runoff as a potential explanatory factor in our discussion. The lateral inputs of different nitrate concentrations and isotopic signatures would be accounted for in those measurements. We used model output from the Chesapeake Bay program (using HSPF, a hydrological surface water runoff model that is used to compute TMDLs for this system) to constrain nutrient inputs from the watershed associated with each "box" of the mass balance box model. This is described in the paragraph on line 275-289 and we have further clarified that sentence to read "freshwater and N inputs from the land".

-290 – I did not like the assumption that these other treatment plants would have little impact. An additional 32% is significant and depending upon where it is added could be very important. The locations of the plants are not given but could this account for the lack of change further downsteam post treatment change (line 367)? The authors also don't mention what types of plant these other WTP are (secondary or tertiary). Some secondary plants get to very high values 15N values if there is extensive open aeration. I agree that the net impact of all of these plants will probably be to underestimate biological assimilation but it would seem to be beneficial to constrain the system to the extent possible. Instead, it is dismissed here and then brought up in the discussion (526) where we find out that the total flows are nearly as large as the Blue Plains plant. It is then brought up again on line 674-5. I believe the authors will have a more robust

story if these plants are incorporated into the model.

We modified the first sentence of this paragraph to say "There are 14 other WWTPs dispersed along the estuary below Blue Plains. One of these WWTPs has tertiary treatment (in addition to Blue Plains), four have secondary treatment, and the rest have primary treatment. These other WWTPs have a combined TN load that is 32% of the TN load from Blue Plains. While the loads from these WWTPs are indirectly accounted for in the box model due to their impact on the concentrations in the estuarine water, it was not feasible to directly incorporate the loads from each WWTP into the box model estimates. However, we can assume that the estimated decline in nitrogen loads from the Blue Plains wastewater treatment plant to the mouth of the Potomac River Estuary are conservative estimates and we can assume that the loads from the 14 other WWTPs have little effect on the nitrate isotope signal. First, the additional load from the other WWTPs only adds to the loads estimated further down estuary and thus the measured loss in N load from the Blue Plains wastewater load down-estuary (the difference between the loads at the mouth and at the head of the estuary) is a conservative estimate because it is less then would be expected and consequently underestimates biological assimilation and removal." And then the paragraph goes on to describe how the other WWTPs have little impact on the isotope signal. Overall, our point here is not that the other WWTPs are not significant, but that it was not feasible to include them in the box modeling effort.

We wish we could have included the WWTPs in the mixing model, but data for parameterizing their inputs was not available – especially in terms of isotopic signatures. It would be possible to do some calculations using the box model with hypothetical isotope signatures from the various WWTPs, but if this is required by the reviewer for acceptance of this manuscript we will need to request additional time for revisions.

The lack of change further down-estuary (noted on line 367) was actually attributed to the 14 other WWTPs on line 525, but the discussion is being revised to further clarify this and other sections of the discussion (as suggested in the comments below).

Because we don't accurately know the nitrate isotope signal from the 14 other WWTPs it is not feasible to incorporate their impact on the isotope mixing model directly. Also, based on the new discussion text referred to above for this same question, we assumed the other WWTPs had little impact on the isotope signal compared to the Blue Plains WWTP. We also added text to this section saying "It is reasonable to say that because 13 of the 14 other WWTPs have only primary or secondary treatment that the nitrate signal isotope values are going to be lower compared to Blue Plain due to the lack of isotopic fractionation from denitrification."

We also acknowledge the potential uncertainties in these assumptions within the discussion.

-414 – Fractionation will only be apparent if only part of the pool is used. While this would seem to be the case, because, nitrate does not completely disappear, there is data from a variety of sources that indicates that sometimes denitrification occurs in hot spots (like hyporheic zones) where part of the pool is completely denitrified without any change in the isotopic composition. I think this at least deserves some mention.

I added this information to the discussion.

-458-475 - I think this could be made clearer. I was initially quite concerned about the very large error bars. The authors attribute some of this to the uncertainties in the last box but in looking at Table 1 things don't improve that much when box 6 is omitted. If I assume all of the seasons are of equal length (3 months) than the seasonal averages presented on 458-460 work out to a loss of 9.03x106 kg/year. With the propagated error this is nearly +/- 100%! But this can be compared to the independent estimates of burial and denitrification rates presented in Boynton et al. 1995 (lines 469-474) of 9.89 x106 kg/year. This agreement is quite good, and I wonder if these huge error bars are due to the method of error propagation. A monte carlo approach might result in smaller errors. I think I would point out the good agreement before going on to attribute the % loss to burial and denitrification.

When I propagate the error I find that on an annual average 9.1 x 106 ± 5.1 x 106 kg/yr of TN are exported to the Bay. I added this to the results and discussion and said that this is a close to Boynton et al. (1995) who estimated 14.1 x106 kg/year are exported from the Potomac River (I did not see the 9.89 x106 kg/year value referred to in this comment).

Should the Monte Carlo error propagation be deemed critical for acceptance, we will need additional time. We have attempted to explore this, but with the multiple uncertainties in parameterization of the box model this has not been immediately obvious to constrain. Estimating errors in box models is a very new approach to take, in fact carrying out the error propagation efforts described here has been a point of discussion with many of our colleagues who regularly use box models (but rarely estimate uncertainty). We expect that a future manuscript will delve into these methods with greater attention.

-Section 4.2 and 4.3 This discussion is quite long and discusses many possible explanations for some of the data but seldom comes to strong conclusions. The authors have some great data here, I'm not sure they are making the most of it. These section contains a lot of statements such as those on line 607 "15N-NO3 values were likely higher in warmer months due to denitrification" since monthly measurements were made don't you know whether or not this is true? No mention is made on line 198 of seasonal changes so I had assumed this was not true. If it is true, the model should be run with different values for different seasons correct? The isotopes are not sufficient to tell when nitrate removal is due to assimilation or denitrification but doesn't the Boynton et al. 1995 data provide some insights that could be used? As mentioned above, on lines 674 the possible role of N from additional treatment plants is brought up when it had been dismissed earlier. So, overall, I think the authors may be able to constrain this system better and come up with more robust conclusions.

For line 607 we removed the word "likely" and instead said that the 15N values were higher "due likely to higher denitrification", because we do know that 15N-NO3 values were higher in the warmer months (nitrate isotopes were measured monthly) but we

are not certain that denitrification is the only cause.

A manuscript is in preparation that leverages this work and that of Cornwell et al. (2016) to establish nitrogen budgets for the Potomac estuary. That is the publication we are planning to use for exploration of the biogeochemistry of these rates in greater detail and where detailed flux measurements will be used to estimate assimilation, etc. That effort also included primary productivity estimates and N efficiency rates that we think will better inform the reviewer's points here. In particular, we will better be able to estimate the proportion of burial, denitrification, and assimilation. Stay tuned!

In the discussion we have acknowledge that it would be helpful to develop seasonal isotope mixing models due to our results showing that temperature and seasonality play a role. But due to lack of data on the seasonality of fertilizer and nitrification endmembers we do not think it is feasible for the scope of this paper.

We have changed the wording to the previous section about the 14 other WWTPs so that we are not dismissing the fact that they contribute a significant load or volume of water, but that their contribution does not adversely impact the trends.

-Conclusions – the importance of hydrology and temperature in N transformation is a critical issue for management and often discussed but removal is also a function of total load. I agree with all of the authors statements but differences in the N behavior in the manuscript is largely discussed by season and I think the conclusions could do a better job talking about all three factors.

I have added in further information for annual averages into the results and discussion section.

———————————————

---

## Author Comment (AC2) · 7 Oct 2016

Please note that my response included a response to both referee 1 and referee 2.

Thank you, Michael Pennino